# All-Trans Retinoic Acid Stimulates Viral Mimicry, Interferon Responses and Antigen Presentation in Breast-Cancer Cells

**DOI:** 10.3390/cancers12051169

**Published:** 2020-05-06

**Authors:** Marco Bolis, Gabriela Paroni, Maddalena Fratelli, Arianna Vallerga, Luca Guarrera, Adriana Zanetti, Mami Kurosaki, Silvio Ken Garattini, Maurizio Gianni’, Monica Lupi, Linda Pattini, Maria Monica Barzago, Mineko Terao, Enrico Garattini

**Affiliations:** 1Laboratory of Molecular Biology, Istituto di Ricerche Farmacologiche Mario Negri IRCCS, via Mario Negri 2, 20156 Milano, Italy; marco.bolis@marionegri.it (M.B.); gabriela.paroni@marionegri.it (G.P.); maddalena.fratelli@marionegri.it (M.F.); arianna.vallerga@guest.marionegri.it (A.V.); luca.guarrera@guest.marionegri.it (L.G.); adriana.zanetti@marionegri.it (A.Z.); mami.kurosaki@marionegri.it (M.K.); maurizio.gianni@marionegri.it (M.G.); mariamonica.barzago@marionegri.it (M.M.B.); mineko.terao@marionegri.it (M.T.); 2Functional Cancer Genomics Laboratory, Institute of Oncology Research, USI, University of Southern Switzerland, 6500 Bellinzona, Switzerland; 3Bioinformatics Core Unit Institute of Oncology Research, Swiss Institute of Bioinformatics, 1000 Lausanne, Switzerland; 4Department of Oncology, Azienda Ospedaliera di Udine, DAME, Dipartimento di Area Medica Università degli Studi di Udine, 33100 Udine, Italy; silvioken@hotmail.it; 5Department of Oncology, Istituto di Ricerche Farmacologiche Mario Negri IRCCS, via La Masa 19, 20156 Milano, Italy; monica.lupi@marionegri.it; 6Department of Electronics, Information and Bioengineering, Politecnico di Milano, 20156 Milano, Italy; linda.pattini@polimi.it

**Keywords:** breast cancer, retinoic acid, antigen presentation, interferon, immune response

## Abstract

All-trans retinoic acid (ATRA), a recognized differentiating agent, has significant potential in the personalized/stratified treatment of breast cancer. The present study reports on the molecular mechanisms underlying the anti-tumor activity of ATRA in breast cancer. The work is based on transcriptomic experiments performed on ATRA-treated breast cancer cell-lines, short-term tissue cultures of patient-derived mammary-tumors and a xenograft model. ATRA upregulates gene networks involved in interferon-responses, immune-modulation and antigen-presentation in retinoid-sensitive cells and tumors characterized by poor immunogenicity. ATRA-dependent upregulation of these gene networks is caused by a viral mimicry process, involving the activation of endogenous retroviruses. ATRA induces a non-canonical type of viral mimicry, which results in increased expression of the *IRF1* (Interferon Responsive Factor 1) transcription factor and the DTX3L (Deltex-E3-Ubiquitin-Ligase-3L) downstream effector. Functional knockdown studies indicate that *IRF1* and *DTX3L* are part of a negative feedback loop controlling ATRA-dependent growth inhibition of breast cancer cells. The study is of relevance from a clinical/therapeutic perspective. In fact, ATRA stimulates processes controlling the sensitivity to immuno-modulatory drugs, such as immune-checkpoint-inhibitors. This suggests that ATRA and immunotherapeutic agents represent rational combinations for the personalized treatment of breast cancer. Remarkably, ATRA-sensitivity seems to be relatively high in immune-cold mammary tumors, which are generally resistant to immunotherapy.

## 1. Introduction

All-trans retinoic acid (ATRA), the active metabolite of vitamin A, is used in the treatment of acute promyelocytic leukemia, with remarkable results [1,2]. ATRA is a non-conventional anti-tumor drug, being the first example of a cyto-differentiating agent [3]. The biological/pharmacological activity of ATRA is mediated by the RAR and RXR nuclear receptors, which are ligand-activated transcription factors controlling the activity of numerous genes, either directly or indirectly [4]. RARs and RXRs work as heterodimerimeric complexes, in which RARs act as the ligand binding-moieties. ATRA is a pan-RAR agonist, binding and activating RARα, RARβ and RARγ with the same efficiency.

A large mass of pre-clinical data indicates that ATRA is a promising agent in the prevention/treatment of breast cancer [5,6,7]. Breast cancer (BC) is a very heterogeneous disease [8] which is traditionally classified in three major groups for clinical purposes, i.e., estrogen-receptor-positive (*ER^+^*), HER2-positive (*HER2^+^*) and triple-negative (*TNBC*). Heterogeneity is a major problem for the rational therapeutic use of ATRA in mammary carcinoma. The problem must be solved with a personalized/stratified approach to therapy which requires the identification of the tumor groups and individual cases sensitive to the retinoid. In previous studies, we demonstrated that approximately 70% of *ER^+^* mammary tumors are sensitive to the anti-proliferative effects of ATRA, while only 10–20% of the *HER2^+^* and *TNBC* counterparts respond to the retinoid [9,10]. In addition, we demonstrated that the anti-proliferative action exerted by ATRA in breast cancer cells is mediated by RARα [9]. However, RARα is a necessary, though insufficient, determinant of ATRA growth-inhibitory activity and its expression does not predict sensitivity to the retinoid [9]. This led us to develop a model consisting of 21 genes (*ATRA-21*) which correctly predicts ATRA-sensitivity in the context of breast cancer and other tumor types [10]. In the context of personalized medicine, the model is likely to represent a useful diagnostic tool for the selection of breast cancer patients who may benefit from ATRA-based therapeutic strategies.

The discovery that tumors tend to escape immune surveillance modulating immune checkpoints [11] has led to the renaissance of immunotherapy for the personalized treatment of different types of tumors [12,13]. Indeed, immuno-therapeutic strategies based on the use of pharmacological agents, such as checkpoint inhibitors targeting the PDL1/PD1 and CTLA4 pathways, represent major advances in cancer treatment. However, only a small percentage of patients are fully responsive to this type of treatment, and this is particularly true in the case of breast cancer [14]. With respect to this, major clinical needs are the identification of response biomarkers and the development of combinatorial treatments enhancing the efficacy of immuno-therapy [15]. One of the main determinants of the response to immune checkpoint inhibitors is tumor immunogenicity, which is the result of neo-antigen expression and antigen-presentation efficiency [16]. Thus, activation of these biological processes is likely to improve the anti-tumor potential of immune-therapeutics and to increase the fraction of cancer patients responding to these agents.

In this study, we use an RNA-sequencing (RNA-seq) approach to define the perturbations afforded by ATRA on the gene-expression profiles of breast cancer, with the use of 8 luminal and 8 basal cell-lines showing variable sensitivity to the anti-proliferative action of ATRA and recapitulating the heterogeneity of the tumor. The results were confirmed in patient-derived short-term tissue cultures and a xenograft model. The data obtained gave insights into the molecular mechanisms underlying the anti-proliferative activity of ATRA in breast cancer. In retinoid-sensitive cell-lines and tumors, ATRA stimulated a viral-mimicry response involving the activation of the interferon (IFN) and antigen-presentation pathways as well as other biological processes controlling the sensitivity of tumors to immune checkpoint inhibitors. In particular, activation of a viral-mimicry response resulted in the induction and activation of Interferon Responsive Factor 1 (*IRF1*) and Deltex-E3-Ubiquitin-Ligase-3L (*DTX3L*). Our functional data indicated that *IRF1* and *DTX3L* exert opposite effects on ATRA-dependent growth inhibition of breast cancer cells, suggesting that they are part of a negative feedback loop. From a therapeutic perspective, the work provides proof-of-principle that ATRA and immunotherapeutic agents represent novel and rational combinations to be tested in the personalized treatment of breast cancer.

## 2. Results

### 2.1. ATRA Upregulates Gene Sets Controlling Interferon/Immune-Modulatory Responses and Antigen-Presentation in Breast Cancer Cell-Lines

In previous studies, we profiled over 50 breast cancer cell-lines for their sensitivity to the anti-proliferative effects of ATRA, using a quantitative index which we denominated “*ATRA-score*” [9,10]. To define the gene sets involved in the growth-inhibitory action of ATRA, we selected a representative panel of 16 cell-lines balanced for the luminal/basal phenotype and for the sensitivity to ATRA (Figure 1A). Indeed, 8 cell-lines have a luminal phenotype, while the other 8 show a basal phenotype and can be further sub-grouped into triple-negative breast cancer (*TNBC*) and triple-negative-mesenchymal (*TNBC-mes*), according to their morphological characteristics and constitutive gene-expression profiles. We re-assessed the sensitivity of the 16 breast cancer cell-lines to the anti-proliferative effects of ATRA, using a recently developed and optimized version of the *ATRA-score* [9,10] (see the Materials and Methods Section). Four luminal cell-lines (*SK-BR-3*/*HCC-1500*/*CAMA-1*/*MDA-MB-361*) are highly sensitive to ATRA, while the other ones (*HCC-202*/*MDA-MB-175VII*/*ZR.751*/*HCC-1419*) show relative resistance to the anti-proliferative effects of the retinoid (Figure 1B). Consistent with the observed higher ATRA-sensitivity of luminal cell-lines relative to the basal counterparts [9], *SK-BR-3*, *HCC-1500*, *CAMA-1* and *MDA-MB-361* cells cluster into the high-sensitivity group, while *HCC-202*, *MDA-MB-175VII*, *ZR.751* and *HCC-1419* cells cluster into the intermediate sensitivity group. As for the basal counterparts (Figure 1B), 4 cell-lines (*HCC-1599*/*MB-157*/*MDA-MB-157*/*Hs578T*) can be classified as responsive to ATRA, while the remaining four cell lines (*HCC-1187*/*CAL-851*/*MDA-MB-436*/*MDA-MB-231*) are highly resistant to the retinoid. Indeed, *HCC-1599* cells are endowed with the highest *ATRA-score* value of the entire panel, while the *ATRA-score* values aggregate *MB-157*, *MDA-MB-157* and *HS578T* cells into the intermediate sensitivity group (Figure 1B). In line with the observed resistance to ATRA, the *ATRA-score* values of *HCC-1187*, *CAL-851*, *MDA-MB-436* and *MDA-MB-231* cells assemble them into the low-sensitivity group. No association is observed between ATRA-sensitivity and the *TNBC* or *TNBC-mes* phenotype of the 8 basal cell-lines. In fact, two *TNBC* (*HCC-1599*/*MB-157*) and two *TNBC-mes* cell-lines (*MDA-MB-157*/*HS578T*) are sensitive two ATRA. By the same token, two *TNBC* (*CAL-851*/*HCC-187*) and two *TNBC-mes* cell-lines (*MDA-MB-231*/*MDA-MB-436*) show resistance to the anti-proliferative effects of the retinoid.

To determine the perturbations afforded by ATRA on gene-expression, we performed RNA-sequencing (*RNA-seq*) studies on these 16 cell-lines exposed to ATRA (1 µM) or vehicle (DMSO) for 24 h. Principal component analysis (PCA) of the transcriptomic data obtained under standard growth conditions separates the luminal from the basal cell-lines (DMSO treatment). As expected, basal cell-lines cluster in the two distinct *TNBC* and *TNBC-mes* sub-groups, reflecting the histochemical and morphological characteristics of the single cell types (Appendix A). ATRA treatment does not cause transitions across the 3 groups, although the retinoid up- and downregulates several genes in each cell-line (Appendix A).

Following application of several filters (Appendix A), we identified 754 genes (upregulated = 340, downregulated = 414) whose expression changes are linearly correlated to the *ATRA-scores* of each cell-line (Appendix A). The results were validated by RT-PCR experiments performed on 4 selected genes (Appendix A). The 754 genes were subjected to pathway-enrichment analysis using different approaches. Initially, we constructed a protein-interaction network with the STRING database, identifying one complex downregulated module controlling cell-cycle/DNA-repair/chromatin-structure and one upregulated module controlling immuno-modulatory/interferon-responses/antigen-presentation (Figure 2). Downregulation of the DNA-repair genes suggests that at least part of the ATRA-dependent growth-inhibitory effect results from a retinoid-triggered genome-instability phenotype [17]. 

Subsequently, we performed Gene Set Enrichment Analysis (GSEA) of the HALLMARK collection using the entire set of genes pre-ranked for their significance (Appendix A and Figure 3A). In retinoid-sensitive luminal and basal cell-lines, ATRA downregulates the “*Myc-Targets*”, “*E2F-Targets*” and “*G2M-checkpoint*” gene sets (Figure 3A and Appendix A), consistent with the anti-proliferative and the known anti-MYC actions of ATRA [18,19]. In the same cell-lines, “*Interferon Alpha Response*” and “*Interferon Gamma Response*” are the top enriched gene sets showing upregulation (Figure 3A and Appendix A). Similar types of analyses performed with the CURATED gene sets (Appendix A) show that some of the top gene sets downregulated by ATRA in sensitive cell-lines are involved in the control of proliferation/cell-cycle, histone/chromatin organization and the MYC-pathway. Overall, these results reinforce the observations made with the STRING/HALLMARK analyses.

To confirm and extend the data described above, we took a less stringent approach and identified the genes up- and downregulated by ATRA in cell-lines characterized by low, intermediate and high retinoid-sensitivity (Appendix A). In the group of cell-lines showing low-sensitivity, ATRA causes significant modifications in the expression of 1110 genes (upregulated = 555, downregulated = 555; adjusted *p*-value < 0.05, following correction with the Benjamini–Hochberg method). In intermediate- and high-sensitivity cell-lines, the retinoid upregulates 2844 and 3151 mRNAs, while it downregulates 3296 and 2938 mRNAs, respectively (adjusted *p*-value < 0.05). GSEA performed with the HALLMARK gene set collection demonstrates enrichment of the pathways listed in Appendix A. For each cell-line group, the top pathways showing a threshold FDR (False Discovery Rate) value > 0.05 are illustrated in Appendix A. As expected, ATRA causes a significant downregulation of the gene networks involved in cell proliferation (“*E2F-Targets*” and “*G2M-checkpoint*”) and MYC activity (“*Myc-Targets*”) only in cell-lines characterized by intermediate- and high-sensitivity to the retinoid. With respect to this last point, it is interesting to notice that ATRA upregulates the “*Myc-Targets*” gene set in the cell-lines characterized by low retinoid sensitivity, supporting the idea that an anti-MYC action underlies the growth-inhibitory effects of the retinoid. By the same token, ATRA-dependent upregulation of the “*Interferon Alpha Response*” and “*Interferon Gamma Response*” gene sets is observed in the cell-line groups showing intermediate and high retinoid-sensitivity. Overall, these results confirm the data illustrated in Figure 3A.

To evaluate whether ATRA exerts similar types of effects in vivo, we analyzed the gene-expression results obtained from tumor tissues of mice transplanted with *HCC-1599* cells and treated with the retinoid [9]. In this experimental model, “*Interferon Alpha Response*” and “*Interferon Gamma Response*” are the 2 top HALLMARK gene sets upregulated by ATRA (Figure 3B and Appendix A). Conversely, “*Myc-Targets-V2*” and “*E2F-Targets*” are among the top downregulated HALLMARK gene sets. Analysis of the CURATED data (Appendix A) shows up- and downregulation of similar pathways. Hence, the transcriptional effects of ATRA are observed not only in vitro but also in vivo. 

### 2.2. ATRA Increases Antigen-Presentation and Interferon/Immune-Modulatory Responses in Cells and Tumors with Low Immunogenicity

Given its potential therapeutic relevance, we focused our attention on the antigen-presentation pathways stimulated by ATRA. Indeed, several gene sets controlling antigen presentation are significantly upregulated by ATRA in retinoid-sensitive cells (Appendix A, CURATED gene sets). For instance, the reactome pathway, *“Antigen-Presentation Folding Assembly and Peptide-Loading of Class-I MHC”* (*“MHC1-APPL”*), consists of 20 genes and it appears among the top enriched pathways upregulated by the retinoid in sensitive cell-lines, regardless of the luminal or basal phenotype (Figure 4A). A similar ATRA-dependent enrichment of the *“MHC1-APPL”* gene set is also observed in *HCC-1599* xenografts (Figure 4B).

GSEA of the *RNA-seq* data obtained in 47 breast cancer cell-lines (Appendix A) [9] grown under standard conditions indicates that constitutive expression of the *“MHC1-APPL”* and *“IFNα-Response”* gene sets is low in highly sensitive cell-lines (*ATRA-score* > 0.17) and high in their counterparts characterized by low ATRA-sensitivity (*ATRA-score* < 0.17) (Figure 4C). Differences between ATRA-responsive and ATRA-unresponsive cell-lines are not observed if the *“Fatty acid metabolism”* gene set (negative control) is considered. This suggests that ATRA-sensitivity is relatively high in cell-lines with low constitutive immunogenicity, which is a characteristic of “immune-cold” tumors [16,20].

To confirm the data, we performed microarray-based transcriptomic studies in patient-derived breast cancer tissue slices exposed to ATRA or vehicle for 48 h [5]. We selected 9 “immune-cold” samples characterized by low constitutive expression of gene networks regulating interferon/immune-modulatory responses. In our experimental conditions, 6 samples respond to ATRA with a decrease in the levels of the cell-proliferation marker, Ki-67 (Figure 4D). Thus, the group is enriched for tumors responsive to the retinoid-dependent anti-proliferative action. In these tumors, ATRA upregulates the *“MHC1-APPL”* and the *“IFNα-Response”* gene sets, as indicated by the heatmaps shown in Figure 4D. This confirms that ATRA stimulates antigen-presentation in the majority of retinoid-sensitive and immune-cold breast cancer tissues. Thus, immune-cold mammary tumors seem to be particularly sensitive to ATRA, which is consistent with what is observed in breast cancer cell-lines.

To support the hypothesis, we used the *RNA-seq* data of the TCGA breast cancer cases. We defined the constitutive expression of the *“MHC1-APPL”*/“*IFNα-Response”* gene sets with the *ssGSEA-score* and estimated ATRA-sensitivity with the *ATRA-21* score, a prediction model based on 21 genes [9,10]. A significant inverse correlation between the *ssGSEA-score* and *ATRA-21* values was observed for both gene sets (Figure 4E), corroborating the experimental results obtained. ATRA-dependent activation of antigen-presentation may have therapeutic implications, as it suggests the use of combinations between the retinoid and immune checkpoint-inhibitors [11].

Immunophenograms are proposed as clinical tools for the selection of patients responsive to immune checkpoint-inhibitors [21]. Hence, we generated immmunophenograms for our 16 cell-lines, using the *RNA-seq* data obtained following exposure to ATRA. As exemplified by the results obtained in the two most sensitive luminal (*SKBR-3*; *HCC-1500*) and basal (*HCC-1599*; *MB-157*) cell-lines, ATRA upregulates various genes involved in antigen-presentation (Figure 5A). In particular, ATRA upregulates PDL1, whose cancer-cell levels are associated with sensitivity to checkpoint-inhibitors in different tumors [22]. The immunophenograms obtained in five of the other luminal cell-lines (*CAMA1*; *MDA-MB-361*; *HCC-202*; *MDA-MB-175VII*; *ZR75.1*; Appendix A) as well as the two other basal and retinoid-sensitive cell-lines (*MDA-MB-157*; *Hs578T*; Appendix A) show upregulation of some MHC and immune check-point genes. This is similar to what is observed in *HCC-1599* and *SKBR-3* cells. The luminal *HCC-1419* and the basal, *MDA-MB-231*, *MDA-MB-436*, *CAL-851*, *HCC-1187* cell-lines, which are characterized by the lowest *ATRA-score* values, do not show significant immune-modulatory responses to ATRA.

Upregulation of the genes involved in MHC class I-mediated antigen-presentation translates into an increase in surface-expression of the corresponding proteins (HLA class I antigens), as verified in *SKBR3* cells (Appendix A). Collectively, our data suggest that combinations of ATRA and immune checkpoint inhibitors have therapeutic potential.

### 2.3. ATRA-Dependent Stimulation of Viral-Mimicry in Cell-Lines

Viral-mimicry is an emerging intra-cellular process involved in the activation of IFN-dependent [23] and other immune responses [24,25] by certain anti-tumor agents, such as demethylating compounds [26,27]. As ATRA-dependent induction of interferon-responsive genes is accompanied by upregulation of gene sets involved in inflammatory/immune responses (see “*Inflammatory-Response*” in Figure 3A), it is possible that ATRA reactivates endogenous retroviruses (*ENVs*) resulting in a viral-mimicry response [28]. To support this hypothesis, we re-analyzed the *RNA-seq* data looking for *ENV*-derived small RNAs. The *RNA-seq* results permit the identification of more than 5 × 10^6^
*ENV*-derived small RNAs which can be grouped in 42 distinct families (Appendix A). Consistent with the hypothesis that ATRA activates viral-mimicry, the *RNA-seq* data demonstrate that the retinoid causes a significant upregulation of various classes of RNAs deriving from *ENV* transposable-elements in different cell-lines. To quantitate the overall effect exerted by ATRA and to correlate this parameter with retinoid sensitivity, we determined the median induction values of the single classes of ENV-derived mRNAs identified in each cell-line (Figure 5B). If these median induction values and the *ATRA-scores* are linearly correlated across our entire panel of cell-lines (Appendix A), the calculated correlation index is low (R^2^ = 0.1629). A close inspection of the diagram suggests that the low R^2^ value is largely explained by the results obtained in luminal cell-lines, which prompted us to perform the same type of analysis following separation of the luminal and basal cell-lines. In luminal cell-lines, no correlation (R^2^ = 0.0043) is observed between the *ATRA-scores* and the median values of the ATRA-dependent induction of these RNAs (Figure 5C). Indeed, *HCC-1419* and *ZR75.1*, which are characterized by the lowest *ATRA-scores*, show a relatively high ATRA-dependent induction of *ENV*-derived RNAs. The situation is different in basal cell-lines, where the two parameters are directly correlated (R^2^ = 0.5788) (Figure 5C). In fact, ATRA upregulates *ENV*-derived RNAs only in the 4 retinoid-sensitive basal cell-lines (Figure 5B). Thus, in the luminal cellular context, the ATRA-triggered induction of *ENV*-derived RNAs and the consequent viral-mimicry response are not associated with sensitivity to the anti-proliferative action of the retinoid. In contrast, the two retinoid-dependent processes are activated only in basal cell-lines characterized by sensitivity to the growth inhibitory effects of ATRA. ATRA-dependent induction of *ENV*-derived RNAs is a relatively early event largely preceding the cell-growth arrest observed in the presence of the retinoid. In fact, increased expression of these RNAs is already observed following 8-h exposure to the retinoid in sensitive *HCC-1599* and *MB-157* basal cell-lines. 

Viral-mimicry upregulates the expression of interferon-stimulated genes and the activation of immune responses via recognition of the *ENV*-derived RNAs and activation of incompletely characterized downstream pathways [29]. Phosphorylation/activation of the TANK-binding-kinase-1 (*TBK1*) and the p65 transcription factor (*p65*) are biomarkers of viral-mimicry laying downstream of *ENV*-derived RNA recognition [30]. Hence, we studied the effect of ATRA on *TBK1* and *p65* phosphorylation in retinoid-sensitive luminal *SKBR-3* cells. In this cell-line (Figure 5D), ATRA stimulates *TBK1* phosphorylation on Ser-172 and activation of the kinase [31]. ATRA-induced *TBK1* phosphorylation starts to be evident at 24 h and it is maintained until 96 h. In addition, the retinoid causes *p65* phosphorylation on Ser-536, which is associated with activation of the transcription factor.

In conclusion, ATRA upregulates numerous *ENV* RNAs which, in turn, trigger an active viral-mimicry response. In our conditions, viral-mimicry activation seems to induce interferon-responsive genes in the absence of interferon production. Indeed, the constitutive levels of type-I/type-II/type-III interferon mRNAs are either null or barely detectable in most of our cell-lines (Appendix A). Moreover, ATRA causes minor increases in the amounts of some interferon mRNAs in few cell-lines and these effects are independent of retinoid-sensitivity/resistance.

### 2.4. IRF-1 and DTX3L Proteins Are Involved in ATRA-Dependent Growth Inhibition of Breast Cancer Cells

To support the functional involvement of viral-mimicry in the anti-proliferative effects of ATRA, we focused our attention on two important members of the interactomic module, *“MHC class I Antigen-Loading and Presentation/Interferon Response”* (Figure 2), i.e., the upstream transcription factor, *IRF1* and the downstream effector protein, *DTX3L*. *IRF1* is an interferon- and retinoid-inducible gene [32,33] which controls the action of ATRA in leukemia and cervical squamous cells [34,35]. ATRA-dependent *IRF1* induction is at the basis of the synergistic interactions between the retinoid and interferons [36]. *DTX3L* is involved in immune, inflammatory and DNA-damage responses [37,38]. Interestingly, *DTX3L* is also a member of the *ATRA-21* gene-expression model and its constitutive expression level is inversely correlated with ATRA-sensitivity [10]. The expression of *IRF1* and *DTX3L* mRNAs is stimulated by ATRA in all luminal and in retinoid-responsive basal cell-lines. Moreover, pharmacological inhibition of the viral-mimicry effector, *TBK1*, with MRT67307, suppresses ATRA-dependent *IRF1*, but not *DTX3L* induction in *SK-BR-3* cells (Figure 5E).

We evaluated the constitutive expression of *IRF1*/*DTX3L* mRNAs and proteins in 41 of our breast cancer cell-lines (Figure 6A). The levels of *IRF1* or *DTX3L* mRNAs were directly correlated with the amounts of the corresponding proteins in all the cell-lines (Appendix A). In addition, the levels of both *IRF1* and *DTX3L* mRNAs as well as proteins were inversely correlated with the *ATRA-score* values (Appendix A). Basal cell-lines express significantly larger quantities of the *IRF1* mRNA and protein than the luminal counterparts (Figure 6B) and a similar trend is also observed with *DTX3L* (Figure 6C). This is in line with the observation that the levels of *IRF1* and *DTX3L* proteins are generally higher in triple-negative relative to *ER^+^* and *HER2^+^* cell-lines. Similar types of constitutive expression profiles of IRF1 and DTX3L mRNAs can be deduced from the RNA-seq results available for breast cancer tissues in the TCGA dataset (Figure 6D,E). Indeed, basal tumors express higher levels of the IRF1 transcript than HER2+ and Luminal A/Luminal B tumors, while no significant difference in constitutive DTX3L mRNA expression is observed across the PAM50 breast cancer groups. The TCGA data indicate that the levels of IRF1 and DTX3L observed in primary tumors are significantly higher than in normal mammary glands (Appendix A). In addition, the expression levels of the two transcripts are quantitatively correlated (Appendix A).

We determined the action of ATRA on the levels of *IRF1*/*DTX3L* proteins and their sub-cellular localization in our original panel of 16 cell-lines. Cell-lines were exposed to vehicle or ATRA (1 µM) for 24 h, before subjecting the derived nuclear and cytosolic extracts to Western blot analysis (Figure 7A). In vehicle-treated cells, *IRF1* is detectable only in the nucleus, while similar amounts of *DTX3L* are measurable in both the nuclear and cytosolic fractions. ATRA induces *IRF1*-protein in the nuclei of all luminal cell-lines, while induction is observed only in retinoid-sensitive basal cell-lines (*HCC-1599*, *MB-157*, *MDA-MB157*, *Hs578T*). Similar profiles of ATRA-dependent induction in the various cell-lines are observed in the case of the *IRF1* mRNA. As for *DTX3L*, ATRA causes detectable increases of the protein levels in two luminal (*SK-BR-3*, *CAMA1*) and three basal (*HCC-1599*, *MB-157*, *MDA-MB-157*) cell-lines endowed with high and intermediate retinoid-sensitivity, which is largely consistent with what is observed at the mRNA level (Figure 7B). Remarkably, the retinoid-dependent induction profiles of *IRF1*-mRNA/-protein and ENV-derived double-stranded RNAs across our panel of cell-lines are correlated (Appendix A). Indeed, *IRF1*-mRNA/-protein induction is observed in all luminal and only in retinoid-sensitive basal cell-lines. As RARα mediates the anti-proliferative action of ATRA in breast cancer cells [9], we exposed *SK-BR-3* cells to ATRA, AM580 (RARα-agonist), UVI2003 (RARβ-agonist) and BMS961 (RARγ-agonist) (Figure 7C). ATRA and AM580 cause a similar induction of *IRF1* and *DTX3L*, while UVI2003 and BMS961 are devoid of any effect. This indicates that RARα mediates ATRA-dependent *IRF1* and *DTX3L* upregulation. *IRF1* and *DTX3L* induction by ATRA is specific, since it is not observed with other anti-tumor agents, as exemplified by the results obtained in *HCC-1599* cells (Figure 7D).

### 2.5. IRF1 and DTX3L Knockdown Affects the Anti-Proliferative Activity of ATRA in Opposite Directions

To obtain direct evidence on the functional involvement of *IRF1* and *DTX3L* in the growth-inhibitory action of ATRA, we silenced the two genes in *SK-BR-3* cells. In the case of *IRF1*, we used two different small interfering RNAs (siRNAs) targeting *IRF1* (*siIRF1(a)*, *siIRF1(b)*) and a scrambled negative control (*siNC*), which were transiently transfected in *SK-BR-3* cells. Transfected cells were incubated with increasing concentrations of ATRA for 4 days. Relative to the *siNC* counterpart, *siIRF1(a)* and *siIRF1(b)* suppress the basal expression of *IRF1* (Figure 7E). In addition, ATRA-dependent *IRF1* induction is substantially reduced by *siIRF1(b)* and nihilated by *siIRF1(a)*. Significantly, *IRF1*-silencing decreases the constitutive levels and the ATRA-dependent upregulation of *DTX3L*. This suggests that the *IRF1* transcription-factor regulates the expression of the *DTX3L* gene. From a functional point of view, *IRF1*-silencing induces resistance to the anti-proliferative action of ATRA. Indeed, relative to the *siNC* counterpart, both *siIRF1(a)* and *siIRF1(b)* decrease ATRA-dependent inhibition of *SK-BR-3* cell-growth, and the effect is observed at all retinoid concentrations. The results indicate that *IRF1* contributes to the anti-proliferative action of ATRA. Consistent with the involvement of RARα in *IRF1* induction by ATRA, *siIRF1(a)* and *siIRF1(b)* reduce the growth-inhibitory effect of AM580 observed in *siNC*-transfected cells (Appendix A). 

In the case of *DTX3L*, we performed silencing studies, using specific shRNAs in *SK-BR-3* cells. The cell-line was stably infected with two lentiviral constructs containing distinct *DTX3L*-targeting shRNAs (*shDTX3L(a)* and *shDTX3L(b)*), a void lentivirus (*shVOID*) and a lentivirus expressing a non-targeted shRNA (*shNC*)). Following antibiotic selection, we isolated four *SK-BR-3* cell populations with stable insertion of each lentivirus DNA. Cells expressing *shDTX3L(a)* and *shDTX3L(b)* show lower constitutive levels of *DTX3L* than the *shVOID* and *shNC* counterparts (Figure 7F, see DTX3L/TUB values). The silencing action of *shDTX3L(a)* and *shDTX3L(b)* is confirmed at the mRNA level, as indicated by the PCR results obtained for the DTX3L transcript, which are shown in Appendix A. In addition, the ATRA-dependent induction of *DTX3L* protein and mRNA observed in *shVOID* and *shNC* cells is suppressed in the *shDTX3L(a)* and *shDTX3L(b)* counterparts (Figure 7F and Appendix A). Relative to what is observed with *IRF1* silencing, *DTX3L* knockdown results in functionally opposite effects. In fact, *DTX3L* knockdown renders *SK-BR-3* cells more responsive rather than more refractory to the anti-proliferative effects of ATRA.

## 3. Discussion

The present study provides insights into the molecular mechanisms underlying the anti-proliferative activity of ATRA in breast cancer. In fact, the *RNA-seq* studies performed on eight luminal and eight basal cell-lines, showing variable sensitivity to the anti-proliferative action of ATRA, indicate that the retinoid activates both the interferon α/β and the interferon γ pathways. The results are confirmed in short-term tissue cultures and the xenograft model considered. Induction of interferon-responsive genes does not seem to be due to increased production/secretion of any type of interferon. In our cellular and patient-derived primary-tumor models, activation of the interferon-response is quantitatively correlated to ATRA-sensitivity. Hence, the stimulated interferon-response is likely to contribute to ATRA growth-inhibitory action.

Our data suggest that the molecular mechanisms underlying the ATRA-dependent induction of interferon-responsive genes are different in retinoid-responsive basal and luminal breast cancer cells. In basal cells, ATRA-dependent activation of the interferon pathway is likely to be directly consequent to induction of a viral-mimicry response [28,39,40], which, in turn, activates the downstream *TBK1*-kinase [25,30]. ATRA-induced viral-mimicry is associated with increased expression of *ENV* double-stranded RNAs. In the basal context, ATRA stimulates *ENV* activity only in retinoid-sensitive cells, as similar effects are not observed in the retinoid-resistant counterpart. In the luminal context, ATRA stimulates viral-mimicry in all the cell-lines considered and the phenomenon lacks quantitative correlations with retinoid-sensitivity. Thus, in luminal cells, viral-mimicry activation is likely to be necessary, albeit insufficient, for the growth-inhibitory action exerted by ATRA. Noticeably, two other classes of anti-tumor agents, i.e., *CDK4/6* inhibitors [41] and DNA-demethylating agents [27], cause viral-mimicry in models of mammary cancer, suggesting a potential role of the *CDK* and DNA-demethylation pathways in the anti-proliferative effects triggered by ATRA. In addition, it is worthwhile mentioning that viral-mimicry increases antigen-presentation [24] and immune-responses [41], two processes which are activated by ATRA in retinoid-sensitive luminal and basal breast cancer cell-lines.

Currently, the mechanisms responsible for the ATRA-dependent re-activation of *ENV*s and the consequent process of viral-mimicry are unknown. In addition, the viral-mimicry response stimulated by ATRA is characterized by some peculiarities. Unlike what is observed with DNA-demethylating agents [27], ATRA-induced viral-mimicry does not increase expression of the *IRF3* and *IRF7* transcription-factors [30], as the only member of the family upregulated by the retinoid is *IRF1*. The absence of *IRF3* and *IRF7* induction may be the reason as to why ATRA treatment upregulates several interferon-responsive genes without an increase in interferon production/secretion [30]. In general, our data are in agreement with the notion that *IRF1* can exert an anti-viral response via interferon-independent upregulation of interferon-stimulated genes. The ATRA-dependent induction profiles of *IRF1* and *ENV*-derived double-stranded RNAs across our panel of cell-lines are correlated. Indeed, *IRF1* induction is observed in all luminal and only in retinoid-sensitive basal cell-lines. This suggests that viral-mimicry is a major determinant of ATRA-dependent *IRF1* induction. Suppression of ATRA-induced *IRF1* upregulation by the TBK1 inhibitor, MRT67307, supports this idea. *IRF1* induction consequent to viral-mimicry activation contributes to the growth-inhibitory action exerted by ATRA, as indicated by the *IRF1*-silencing studies.

Another relevant result regarding the molecular mechanisms underlying the action of ATRA is the identification of a potential negative-feedback loop involving the interferon-dependent *DTX3L* gene, an E3-ubiquitin-ligase [42], which inhibits *IRF1* expression [43]. In *SK-BR-3* cells, *IRF1*-silencing reduces the basal levels and the ATRA-dependent induction of *DTX3L*, in line with the idea that the *DTX3L* gene is an *IRF1* transcriptional target. Consistent with the involvement of *DTX3L* in a negative-feedback loop, silencing of the E3-ubiquitin-ligase sensitizes *SK-BR-3* cells to ATRA growth-inhibitory action, which is the opposite of what is observed following *IRF1*-silencing. In our experimental conditions, *DTX3L*-silencing does not alter RARα expression in *SK-BR-3* cells regardless of ATRA exposure. Interestingly, a recent paper demonstrates upregulation of *DTX3L* upon viral-mimicry stimulation by Poly (I:C) [44].

## 4. Materials and Methods

### 4.1. Cell-Lines

The list, the characteristics and the source of all the cell-lines is available in the Appendix A, Methods Section.

### 4.2. Tissue Slice Cultures of Primary Breast Cancer Specimens

Short-term tissue slice culture experiments were conducted as described in Centritto et al. [5]. The fresh primary tumor samples used for the short-term tissue slice cultures were supplied by Fondazione S. Maugeri, Pavia. The samples were obtained from patients undergoing a Tru-cut diagnostic procedure. All the procedures were approved by the internal ethical committee of the Fondazione S. Maugeri and an informed consent for the donation of the sample was obtained from patients. The relevant clinical characteristics of the 9 patients used in the study are available in the Appendix A, Methods Section (Primary breast cancer specimens).

### 4.3. ATRA-Score

The sensitivity of the cell-lines to the anti-proliferative action of ATRA was determined following optimization of the method necessary to calculate the *ATRA-score* quantitative index [9]. Briefly, breast cancer cell-lines were exposed to vehicle (DMSO) or five logarithmically increasing concentrations of ATRA (0.001–10.0 μM) for 9 days. Cell growth was determined with the sulforhodamine assay [9]. Each experimental point consisted of 6 replicates. For each cell-line, at least two independent experiments were carried out. ATRA-score = log_2_ transformation of the product of AUC X A_max_ (Area Under the Curve x Maximal Area) rescaled in a range between 0 and 1. “0” and “1” indicate total resistance and maximal sensitivity to ATRA, respectively.

### 4.4. IRF1 and DTX3L Silencing Experiments

The following stealth RNAi (Interfering RNA) (Invitrogen-Thermo Fisher Scientific, Carlsbad, CA, USA) were used: *siIRF1(a)* = RNAi (HSS105500) and *siIRF1(b)* = RNAi (HSS179960). Stealth RNAi Negative Control duplexes (high GC, cat. No. 12935-400) were used as negative controls (siNC). Cells were transfected with stealth RNAi oligonucleotides (60 nM) in OptiMem medium containing Lipofectamine 2000 (Invitrogen). To obtain the *DTX3L* silencing construct, the following custom-synthesized double-stranded DNA coding for two different *DTX3L*-targeting shRNAs were used: *shDTX3L(a)* (sense, 5’- GATCCGGAGAAAGGAGGCGAATTACTTCCTGTCAGATAATTCGCCTCCTTTCTCCTTTTTG-3’; antisense, 5’-AATTCAAAAAGGAGAAAGGAGGCGAATTATCTGACAGGAAGTAATTCGCCTCCTTTCTCCG-3’, starting at position 1094 in NM_138287); *shDTX3L(b)* (sense, 5’-GATCCCAGATGTCATCACTTGGAATGATATCTTCCTGTCAGAATATCATTCCAAGTGATGACATCTGTTTTTG-3’; antisense, 5’-AATTCAAAAACAGATGTCATCACTTGGAATGATATTCTGACAGGAAGATATCATTCCAAGTGATGACATCTGG-3’, starting at position 2182 in NM_138287), and were introduced into the pGreenPuro plasmid (System Biosciences Inc., http://www.systembio.com), using the *EcoRI* and *BamHI* sites in the multiple cloning region downstream of the H1 gene promoter. As a negative control, the following sequences were used: *shNC* (5’-GATCCGCGCGATAGCGCTAATAATCTTCCTGTCAGA ATTATTAGCGCTATCGCGC TTTTTG-3’; antisense, 5’-AATTCAAAAAGCGCGATAGCGCTAATAATTCTGACAGGAAG ATTATTAGCGCTATCGCGCG-3’).

The shRNA lentiviral constructs were transfected into *HEK-293* cells to generate the retroviral particles. The particles were infected in *SKBR-3* cells and the cells were selected for resistance to puromycin.

### 4.5. Western Blots and FACS (Fluorescence Activated Cell Sorting) Analysis

Western blot experiments were conducted according to routine protocols [45], with anti-IRF1(Cell Signaling Technology, Danvers, MA, USA, #8478, dilution 1–1000), anti-DTX3L (Cell Signaling Technology, #14795, dilution 1–1000), anti-tubulin (MERCK, Darmstadt, Germany, #T5168, dilution 1–10,000), anti-TBK1 (Cell Signaling Technology, #3505, dilution 1–1000), anti-TBK1 phosphorylated on Serine/172 (Cell Signaling Technology, #5483, dilution 1–1000), anti-p65 (Cell Signaling Technology, #8242, dilution 1–1000), anti-p65 phosphorylated on Serine/536 (Cell Signaling Technology, #3033, dilution 1–1000) primary antibodies and appropriate fluorescence-tagged secondary antibodies [Cy3-conjugated AffiniPure goat anti-mouse IgG (H + L), #115-165-003 and Cy5-conjugated AffiniPure goat anti-rabbit IgG (H + L) #111-175-144, Jackson ImmunoResearch, West Grove, PA, dilution 1–2000]. Western blots were developed with a Typhoon FLA 9500 biomolecular imager (GE Healthcare Bio-Sciences AB, Uppsala, Sweden) and densitometric analysis of the bands was conducted with the associated software. Nuclear and cytoplasmic extracts were obtained with the NE-PER™ Nuclear and Cytoplasmic Extraction Reagents (Thermo Fisher Scientific). FACS (fluorescence Activated Cell Sorting) analyses were conducted according to standard protocols with an antibody targeting the HLA class A surface antigen (SIGMA, SAB4700637, dilution 1–500).

### 4.6. Polymerase Chain Reaction

Total RNA was isolated from the indicated cells with the miRNeasy Minikit (Qiagen, Waltham MA, USA) and the cDNA was synthesized and amplified using the GeneAmp RNA PCR kit (Applied Biosystems, Foster City, CA, USA). The expression of the following transcripts was evaluated with the use of commercially available Taqman gene expression assays according to the instructions of the manufacturer (ThermoFisher Scientific): *IRF1* (Hs00233698_m1; interrogated sequence-NM_002198.2;) and *DTX3L* (Hs00370540_m1; interrogated sequence-NM_138287.3). The internal control used for the normalization of the data is the 18S RNA, which was determined with the use of the standard Taqman gene expression assay, Hs99999901_s1 (ThermoFisher Scientific). Real Time PCR was performed with the following instrument: 7300 Real Time PCR system (Applied Biosystems).

### 4.7. RNA-Sequencing Studies

Three paired biological replicates of each breast cancer cell-line were grown in DMEMF12 medium containing 5% charcolated FBS (Fetal Bovine Serum, Gibco, Carlsbad, CA, USA) for 24 h. Cells were treated with vehicle (DMSO, Sigma) or ATRA (1 μM) for another 24 h. RNA was extracted with the mRNeasy Mini Kit (QIAGEN). RNA sequencing was performed using the Illumina TruSeq RNA library preparation kit (Illumina, San Diego, CA, USA) and sequenced on the Illumina NextSeq500 with paired-end, 150 base-pair long reads. The overall quality of sequencing reads was evaluated using FastQC [3]. Sequence alignments to the reference human genome (GRCh38) were performed using STAR (v.2.5.2a). Gene-expression was quantified using the comprehensive annotations available in Gencode [4]. Specifically, we used the v27 release of the Gene Transfer File (GTF). Raw counts were further processed in the R Statistical environment and downstream differential expression analysis was performed using the DESeq2 pipeline. Genes characterized by low mean normalized counts were filtered out by the Independent Filtering feature embedded in DESeq2 (alpha = 0.05). DESeq2-computed statistics were used as input for gene set enrichment testing performed with the pre-ranked version of Camera. Statistical enrichments were determined for gene sets obtained from the HALLMARK (H), which are curated by the Molecular Signature DataBase (MSigDB). The RNA-seq data obtained in the 16 breast cancer cell-lines exposed to vehicle or ATRA and the microarray gene-expression results obtained in the breast cancer tissue samples have been deposited in the EMBL-EBI Arrayexpress database (Accession No: E-MTAB-8408).

### 4.8. Quantification of Transposable Elements

Nearly half of the human genome consists of repetitive elements that are tightly regulated to protect the host genome from destructive consequences associated to their inappropriate reactivation [46]. Both full-length and fragmented copies of these viral genomes have propagated through host genomes to produce repeating instances of their sequences [47]. To quantify the expression of these transposable elements, we retrieved their genomic positions from the *RepeatMasker* database (http://www.repeatmasker.org/). We built a custom GTF file (gene transfer file, GTF) including these annotations, along with those of canonical genes, and quantified their abundance using STAR [48]. To avoid detection of false positives, we discarded all transposable elements that showed any overlap to known gene-associated exons, according to the *Gencode* annotations [49]. Repetitive elements were grouped into 42 distinct families, as reported in Appendix A. Differential expression of repetitive elements was performed using the DESeq pipeline. In order to obtain a single quantification for all repetitive elements, we further grouped all members of the above-mentioned 42 classes into a single meta-gene, which consisted of 5,482,861 elements. Differential expression and statistics were computed as previously described in Section 4.7.

### 4.9. Statistics

If not otherwise indicated, statistical analyses were performed using two-tailed statistical tests and corrected for multiple comparisons, using the Benjamini-Hochberg correction method. The statistical treatments performed on the RNA-seq data are detailed in the appropriate sections.

## 5. Conclusions

This study provided insights into the molecular mechanisms underlying the anti-proliferative activity of ATRA in breast cancer. In our models, the retinoid-dependent induction of viral-mimicry and interferon-responsive genes, which involves induction of the *IRF1* transcription factor, was found to be likely to contribute to ATRA growth-inhibitory action. Besides its significance from a mechanistic point of view, the study is of relevance from a clinical and therapeutic perspective. In fact, ATRA stimulates antigen-presentation, interferon-dependent and immune responses, three processes controlling the sensitivity of tumors to checkpoint inhibitors. This suggests that ATRA and immunotherapeutic agents represent rational combinations to be tested for the personalized treatment of breast cancer. With respect to this, it is remarkable that ATRA-sensitivity seems to be higher in “immunologically-cold” mammary tumors, which are generally resistant to immunotherapy. Exploratory clinical studies are required to evaluate whether combinations between ATRA and immunotherapy are indeed characterized by clinical potential in breast cancer.

## Figures and Tables

**Figure 1 cancers-12-01169-f001:**
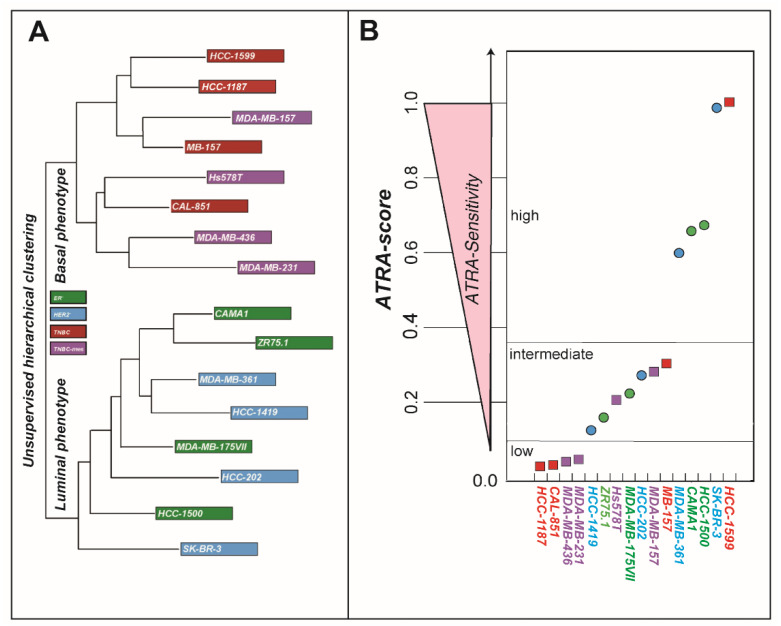
Characteristics and all-trans retinoic acid (ATRA)-sensitivity of the breast cancer cell-lines (**A**) The dendrogram illustrates the unsupervised hierarchical clustering of the indicated 16 breast cancer cell-lines based on the constitutive whole-genome gene-expression profiles determined by RNA-sequencing (*RNA-seq*) analysis. The cell-lines are also classified according to the estrogen receptor, *HER2* receptor (*ERBB2*, erb-b2 receptor tyrosine kinase 2) and morphological characteristics into 4 groups: *ER^+^* = estrogen receptor positive, *HER2^+^* = HER2 positive, *TNBC* = triple-negative breast cancer, *TNBC-mes* = triple-negative breast cancer with a mesenchymal phenotype. (**B**) The indicated cell-lines are ranked according to their sensitivity to the anti-proliferative action of ATRA using the *ATRA-score* index. The higher the *ATRA-score* value, the higher the sensitivity of the cell-line to ATRA. Basal cell-lines are indicated with a square, while luminal cell-lines are indicated with a circle. Cell-lines are classified according to a high, intermediate and low sensitivity to ATRA, as shown.

**Figure 2 cancers-12-01169-f002:**
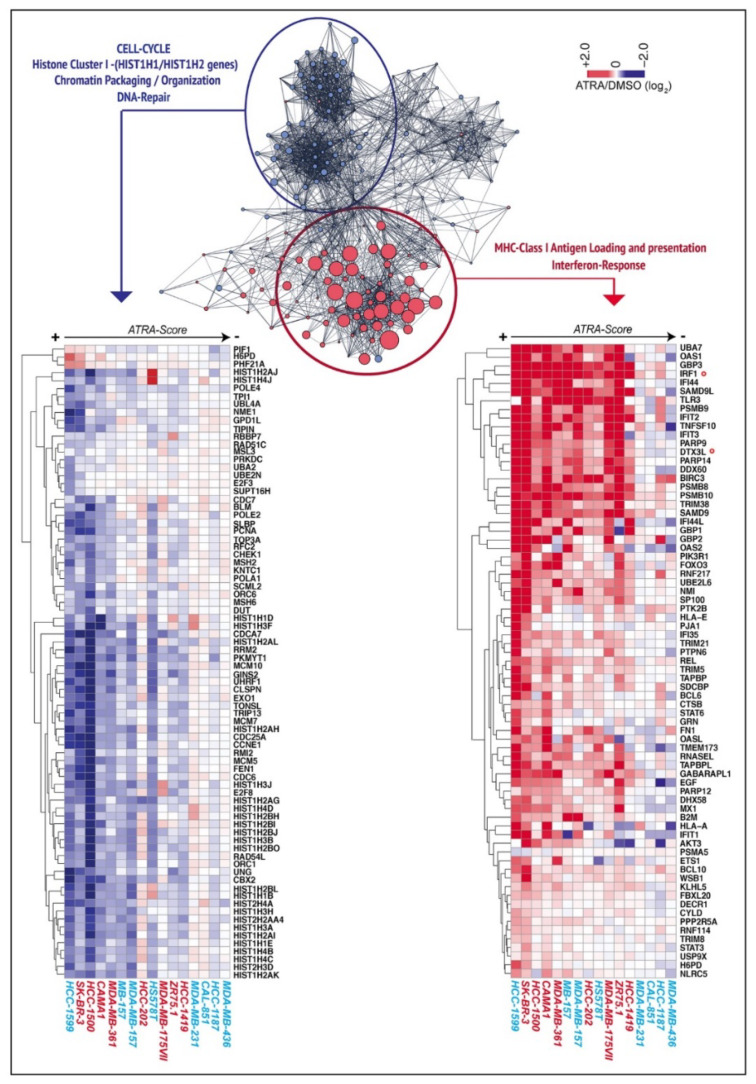
Interaction networks of the genes up- and downregulated by ATRA in the retinoid-sensitive cell-lines. The 754 genes whose up- or downregulation is proportional to ATRA-sensitivity were used to construct an interaction network based on the encoded proteins (STRING database, https://string-db.org). The 2 upregulated modules (*MHC Class-I Loading and presentation*; *Interferon Response*) can be further condensed into a single group, as indicated by a red circle. The single genes belonging to this module are indicated by red dots. The higher the average upregulation afforded by ATRA, the larger the size of each red dot. The 4 downregulated modules (*Cell-cycle*, *Histone-cluster I*, *Chromatin-packaging/organization*, *DNA-repair*) can be further clustered into a single group marked by a blue circle. The single genes belonging to this module are indicated by blue dots. The lower the fold-change observed following ATRA treatment, the larger the size of each blue dot. The heat-maps of the ATRA/DMSO fold changes observed for the genes belonging to the up- (right) and downregulated (left) interaction networks in each cell-line are shown. The cell-lines are ranked from left to right according to their decreasing sensitivity to ATRA (decreasing *ATRA-score* value). Luminal cell-lines are indicated in red, while basal cell-lines are marked in blue. The *IRF1* and *DTX3L* mRNAs are marked with a red circle.

**Figure 3 cancers-12-01169-f003:**
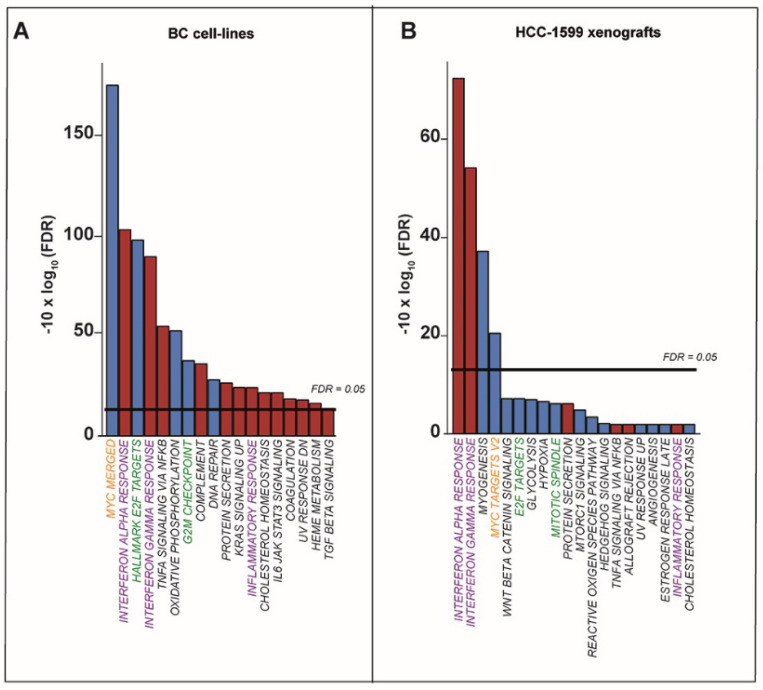
Gene set enrichment analysis of the RNA-seq results obtained in breast cancer cell-lines and *HCC-1599* xenografts. (**A**) Gene set enrichment analysis was performed on the genes whose up- or downregulation is proportional to ATRA-sensitivity using the HALLMARK gene sets. The top 18 enriched gene sets are shown. The blue and red columns indicate the gene sets collectively down- and upregulated by ATRA, respectively. (**B**) *HCC-1599* cells were transplanted subcutaneously in nude mice. Xenografted animals were treated with two daily doses of vehicle or ATRA (15 mg/kg) orally according to the scheme illustrated in [9]. Twenty-four h following the last treatment, the *HCC-1599* xenografts of 3 ATRA-treated and 3 vehicle-treated animals were subjected to whole-genome gene-expression microarray analysis. Genes significantly up- and downregulated by ATRA were subjected to gene set enrichment analysis using the HALLMARK gene sets. The blue and red columns indicate the gene sets collectively down- and upregulated by ATRA, respectively. The horizontal line indicates the FDR (False Discovery Rate) threshold value considered. Green = Gene sets involved in the control of proliferation/cell-cycle; Orange = Gene sets involved in the control of the Myc-pathway; Violet = Gene sets involved in interferon/inflammatory responses.

**Figure 4 cancers-12-01169-f004:**
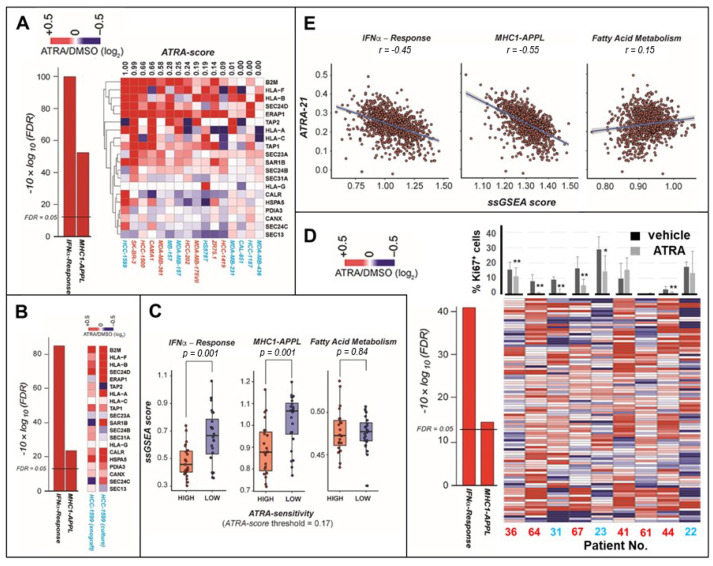
ATRA-related upregulation of antigen-presentation and interferon-α genes in breast cancer. (**A**) The left graph indicates the enrichment of the two indicated gene networks. The horizontal line indicates the FDR (False Discovery Rate) threshold value. The heat-map on the right illustrates the effects of ATRA on the “*MHC1-APPL*” (*Antigen-Presentation Folding Assembly and Peptide-Loading of Class-I MHC*) gene set. The cell-lines are ordered according to the *ATRA-score* from left to right. The *ATRA-score* values are shown above the heat-map. Luminal and basal cell-lines are marked in red and blue, respectively. (**B**) The left graph illustrates the ATRA-dependent enrichment of “*IFNα-Response*” and “*MHC1-APPL*” gene sets in the xenografts of *HCC-1599* cells. The two heat-maps on the right compare the effects of ATRA on the “*MHC1-APPL*” gene set in *HCC-1599* cell-cultures and xenografts. (**C**) Using the CCLE (Cancer Cell-Line Encyclopedia) *RNA-seq* data, the constitutive expression levels of the “*IFNα-Response*”, “*MHC1-APPL*” and *“Fatty Acid Metabolism”* (negative control) gene sets were evaluated using the Single-Sample Gene Set Enrichment Analysis (*ssGSEA-score*) on 47 breast cancer cell-lines (Appendix A). The data are shown following clustering of the cell-lines in two groups characterized by *ATRA-scores* above (HIGH-ATRA-sensitivity) and below (LOW-ATRA-sensitivity) a 0.17 threshold value (Appendix A). Statistical significance was calculated with the pairwise *t*-test corrected for multiple comparisons (*n* = 3), using the Benjamini–Hochberg correction method. (**D**) The panel illustrates the whole-genome gene-expression results of patient-derived tumor tissue slices exposed to ATRA (1 μM) or vehicle (DMSO) for 48 h. The column graph on the left illustrates the ATRA-dependent enrichment of “*IFNα-Response*” and “*MHC1-APPL*” gene sets in the tissue slices. The horizontal line indicates the FDR threshold value. The heatmap on the right illustrates the ATRA-dependent expression of the *“IFNα-Response”* and *“MHC1-APPL”* genes. The response to the anti-proliferative effects of ATRA was determined by measurement of the *Ki-67* proliferation marker in the tissue slices exposed to DMSO/ATRA and the results obtained are illustrated by the column bar-graph shown above the heatmap on the right. ** Significantly different relative to the corresponding DMSO-treated control value (*p* < 0.01 following a two-way *t*-test analysis). * Significantly different relative to the corresponding DMSO-treated control value (*p* < 0.05 following a two-way *t*-test analysis). When tumor cells are characterized by a luminal phenotype, the patient number (No.) is marked in red. Conversely, the patient No. is marked in blue if tumor cells are characterized by a basal phenotype. The following breast cancer subtypes were diagnosed by the pathologist: Patient 36 = Luminal B, Patient 64 = Luminal B, Patient 31 = triple-negative, Patient 67 = HER2-positive (positive for *ERBB2*, erb-b2 receptor tyrosine kinase 2), Patient 23 = triple-negative, Patient 41 = Luminal A, Patient 61 = Luminal B, Patient 44 = Luminal A, Patient 22 = triple-negative. (**E**) Using the *RNA-seq* data of the breast cancer cases of the TCGA dataset, we predicted ATRA-sensitivity using the *ATRA-21* model. The diagrams illustrate the correlations between the *ATRA-21* values and the *ssGSEA-scores*.

**Figure 5 cancers-12-01169-f005:**
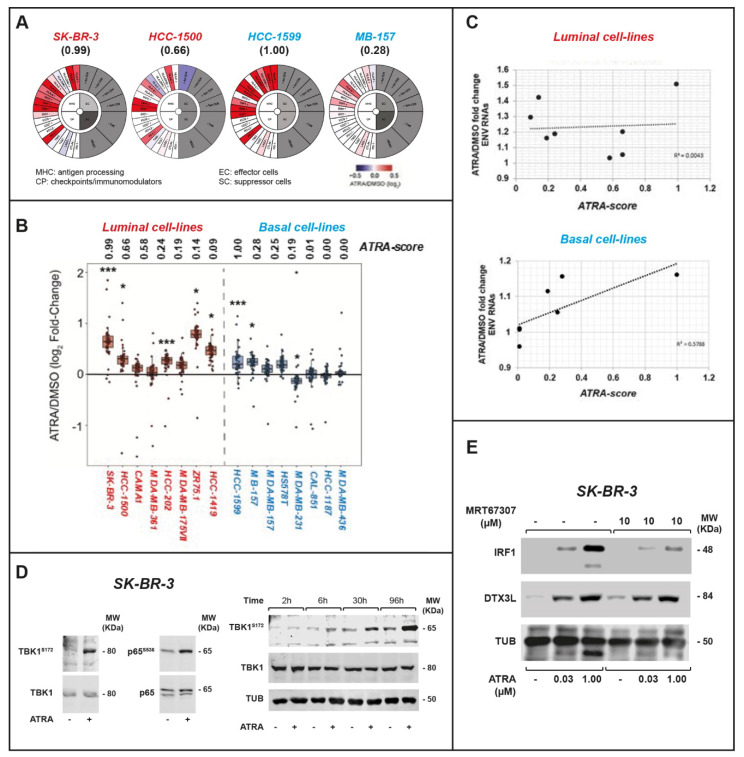
Effects of ATRA on the immunophenograms, endogenous retroviral elements and viral mimicry upstream regulators of breast cancer cell-lines. (**A**) The indicated cell-lines were treated with vehicle (DMSO) or ATRA (1 μM) for 24 h. The circle plots show the effects exerted by ATRA on the immunophenograms derived from the RNA-seq data. The *ATRA-score* values are shown in parenthesis. (**B**,**C**) The indicated breast cancer cell-lines were exposed to vehicle (DMSO) or ATRA (1 μM) for 24 h. (**B**) The box-plots show the effects exerted by ATRA on the expression of individual RNAs deriving from endogenous retroviruses (*ENV*). For each cell-line, the points represent the mean (*n* = 3) ATRA/DMSO fold-change values of the single RNAs determined. Within the luminal and basal groups, cell-lines are positioned according to a decreasing *ATRA-score* from left to right, as indicated. * Significant ATRA-dependent up- or downregulation (*p*-value < 0.1); *** Significant ATRA-dependent upregulation (*p*-value < 0.001). (**C**) The diagrams illustrate the correlations between the effects exerted by ATRA on *ENV*-derived mRNAs and the *ATRA-scores* determined in luminal and basal cell-lines. ATRA/DMSO fold-change *ENV* RNAs = fold-change of the median expression values of the 42 classes of *ENV* RNAs determined in each cell-line following treatment with ATRA (1 μM) or vehicle (DMSO) for 24 h. The median fold-changes are expressed in linear values and were calculated from the logarithmic values shown in (**B**) and Appendix A. The R^2^ values of the correlations are indicated. (**D**) Left: *SK-BR-3* cells were treated with DMSO or ATRA (1 μM) for 24 h. Cell extracts were subjected to Western blot analysis for TBK1, Serine/172 phosphorylated TBK1 (TBK1^S172^), p65 and Serine/536 phosphorylated p65 (p65^S536^). Right: *SK-BR-3* cells were treated with DMSO or ATRA (1 μM) for the indicated amount of time. Total cell extracts were subjected to Western blot analysis for TBK1, TBK1^S172^ and tubulin (TUB) as a loading control. TBK1^S172^/TBK1 = ratio of the densitometric results obtained for the phosphorylated TBK1 and total TBK1 bands (p65^S536^/p65 = ratio of the densitometric results obtained for the phosphorylated p65 and total p65 bands). (**E**) *SK-BR-3* cells were treated with the TBK1 inhibitor, MRT67307, in the presence/absence of ATRA at the indicated concentrations for 24 h. Cells were pre-treated with the TBK1 inhibitor for 1 h. Total cell extracts were subjected to Western blot analysis for IRF1, DTX3L and tubulin (TUB). IRF1/TUB = ratio of the densitometric results obtained for the *IRF1* and Tubulin bands, DTX3L/TUB = ratio of the densitometric results obtained for the *DTX3L* and Tubulin bands. For the whole Western blot figures, please see Appendix A.

**Figure 6 cancers-12-01169-f006:**
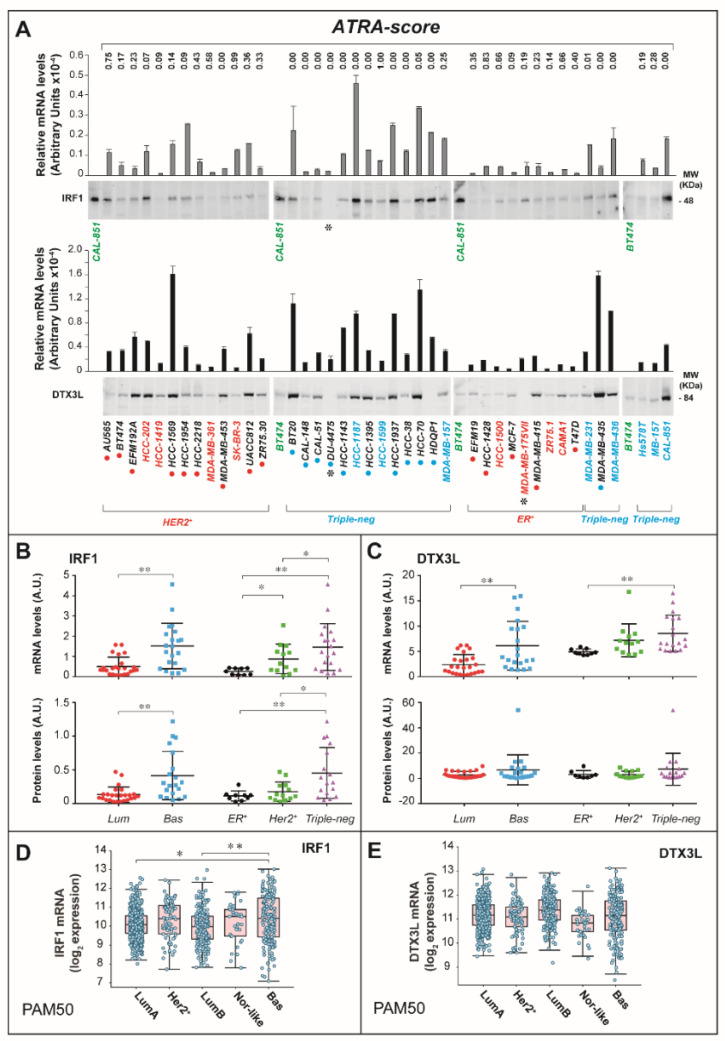
Expression of the IRF1 and DTX3L proteins/mRNAs in breast cancer cell-lines and tissues. (**A**) *IRF1* (Interferon-Responsive-Factor-1) and *DTX3L* (Deltex-E3-Ubiquitin-Ligase-3L) mRNAs were measured by PCR (Polymerase Chain Reaction) in the indicated 41 cell-lines (column graphs). Each value is the mean ± standard deviation (SD) of 3 replicate cultures and the results are normalized for the 18S mRNA. The Western blots illustrate the levels of IRF1 and DTX3L proteins. The 16 cell-lines belonging to our experimental panel are indicated in blue (basal cell-lines) and red (luminal cell-lines). The basal or luminal phenotype of all the other cell-lines is indicated by blue and red dots, respectively. Each well contains an equivalent amount of protein. *CAL-851* and *BT474* cells (internal loading controls) are marked in green. The asterisk underneath the indicated Western blots marks the cell-lines for which bands are not detectable. The positivity of each cell-line to HER2 (*HER2^+^*) or ER (*ER^+^*) as well as the negativity of each cell line to HER2, ER and PR (progesterone receptor) is indicated by the *Triple-neg* (Triple-negative) designation. For the whole Western blot figures, please see Appendix A. (**B**) and (**C**) The *IRF1* as well as *DTX3L* mRNA and protein levels are compared following clustering of the cell-lines according to the luminal (*Lum*) and basal (*Bas*) phenotype or the presence/absence of the *ER* (Estrogen Receptor) and *HER2* proteins as defined in (**A**). Each point represents a single cell-line. The quantitative protein data were obtained from the densitometric analysis of the Western blots shown in (**A**), following normalization with the internal loading controls. * Significantly different following the Student’s *t*-test, corrected for multiple testing (Benjamini–Hochberg) (*p* < 0.05). ** (*p* < 0.01). (**D**) and (**E**) The Box Plots show the *IRF1* and *DTX3L* mRNA levels of breast cancer tissue samples determined from the *RNA-seq* data available in the TCGA database (1108 cases). The breast cancer cases are classified according to the PAM50 algorithm into 5 distinct groups: Luminal A = *LumA*, *HER2*-positive = *HER2^+^*, Luminal B = *LumB*, Normal-like = *Nor-like*, Basal = *Bas*. Each point represents an individual case. * Significantly different following pairwise *t*-test comparison adjusted for multiple testing according to the Benjamini–Hochberg correction method (*p* < 0.05). ** (*p* < 0.01).

**Figure 7 cancers-12-01169-f007:**
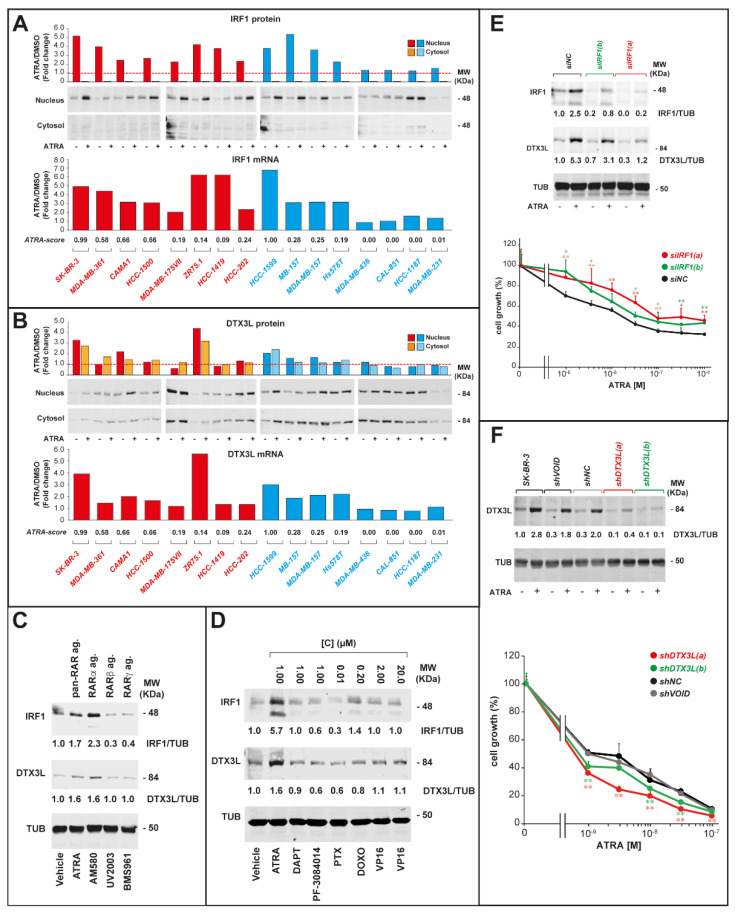
Effects of ATRA and derived retinoids on IRF1 and DTX3L in breast cancer cell-lines and functional significance of the two proteins in ATRA-dependent growth inhibition. (**A**) IRF1 and (**B**) DTX3L protein and mRNA levels were determined in vehicle (DMSO)- or ATRA (1 μM)-treated cell-lines (24 h). IRF1 and DTX3L proteins were determined in the nuclear and cytosolic fraction by Western blot analysis. The column graphs above the Western blots show the densitometric results obtained. The column graphs underneath the Western blots show the effect exerted by ATRA on the expression levels of the *IRF1* and *DTX3L* mRNAs (Taqman assays). (**C**) *SK-BR-3* and (**D**) *HCC-1599* cells were treated with vehicle and the indicated RAR agonist (1 μM), the γ-secretase inhibitors, DAPT [(2S)-N-[(3,5-Difluorophenyl)acetyl]-L-alanyl-2-phenyl]glycine-1,1-dimethylethyl-ester] (1 μM) and PF-3084014 [(2S)-2-[[(2S)-6,8-Difluoro-1,2,3,4-tetrahydro-2-naphthalenyl]amino]-N-[1-[2-[(2,2-dimethylpropyl)amino]-1,1-dimethylethyl]-1H-imidazol-4-yl]-pentanamide-dihydrobromide] (1 μM), PTX (paclitaxel, 0.01 μM), DOXO (doxorubicin, 0.2 μM) and VP16 (etoposide, 2.0 and 20 μM) for 24 h. Cellular extracts were subjected to Western blot analysis for IRF1, DTX3L and tubulin (TUB, loading control). IRF1/TUB = ratio of the densitometric results obtained for the IRF1 and Tubulin bands, DTX3L/TUB = ratio of the densitometric results obtained for the DTX3L and Tubulin bands. (**E**) *SK-BR-3* cells were transfected with the small interfering RNAs, siIRF1(a), siIRF1(b), or the scrambled siNC control. 24 h following transfection, cells were treated with vehicle (DMSO) or the indicated concentrations of ATRA for another 96 h. Cell-growth was evaluated with sulforhodamine assays and the results are expressed as the % growth inhibition relative to the corresponding siNC control (lower panel). Values are the mean ± SD of 3 replicate cultures. * significantly different (Student’s *t*-test, *p* < 0.05; red asterisks: siIRF1(a) versus siNC, green asterisks: siIRF1(b) versus siNC). ** significantly different (Student’s *t*-test, *p* < 0.01; red asterisks: siIRF1(a) versus siNC, green asterisks: siIRF1(b) versus siNC). The effect of silencing on the *IRF1* protein was evaluated after 24-h treatment with vehicle or ATRA (1 μM). Cell extracts were subjected to Western blot analysis for *IRF1* and Tubulin (upper panel). IRF1/TUB = ratio of the densitometric results obtained for the *IRF1* and Tubulin bands, DTX3L/TUB = ratio of the densitometric results obtained for the *DTX3L* and Tubulin bands. (**F**) *SK-BR-3* cells were stably infected with shDTX3L(a), shDTX3L(b), the void lentiviral vector (shVOID) and a scrambled shRNA (shNC). Following antibiotic selection and isolation, the indicated cell populations were treated (24 h) with vehicle (DMSO) or ATRA (1 μM). Total cell extracts were subjected to Western blot analysis for DTX3L and TUB (upper panel). DTX3L/TUB = ratio of the densitometric results obtained for the *DTX3L* and Tubulin bands. Cell-growth (lower panel) was evaluated as in (**E**). Each value is the mean ± SD of 3 replicate cultures. *SK-BR-3* = parental cell-line. * significantly different from shNC and shVOID (Student’s *t*-test, *p* < 0.05). ** significantly different from shNC and shVOID (Student’s *t*-test, *p* < 0.01). For the whole Western blot figures, please see Appendix A.

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
