# Peer review of "All-Trans Retinoic Acid Stimulates Viral Mimicry, Interferon Responses and Antigen Presentation in Breast-Cancer Cells"

_cancers, 2020, doi:10.3390/cancers12051169_

Round 1
Reviewer 1 Report
Dear Authors,
Review of the paper “All-trans retinoic acid stimulates viral mimicry, interferon responses and antigen presentation in breast-cancer cells” by Bolis et al. has been completed. This study reports on the 26 molecular mechanisms underlying the anti-tumor activity of ATRA in breast-cancer. This suggests that ATRA and immunotherapeutic agents represent rational combinations for the personalized treatment of breast-cancer.
Broad comments:
Overall, this is a very thoroughly conducted study, with sound methodology and reporting. There are minor queries and suggestions mentioned below.
Comments:
- The introduction in my opinion needs to provide more information and build a rationale for the study conducted.
- Line 52 remove excess spacing
- Query on whether cell lines authentication has been performed on all 16 cell lines
- Can the authors provide a response to why those particular cell lines were chosen for this study, has any work been conducted on primary tumour cells grown from patients?
- Did the authors collect any patient clinicopathological details on collected primary tumour samples, if so it would be interesting to view a summary of patient characteristics?
- Section 4.5 need to provide information on antibody concentrations used for the experiments, overall a more detailed description of this procedures would be beneficial. The same is the case for the description of PCR reaction in the following section. What were the controls?
Author Response
Reviewer 1
Dear Authors,
Review of the paper “All-trans retinoic acid stimulates viral mimicry, interferon responses and antigen presentation in breast-cancer cells” by Bolis et al. has been completed. This study reports on the 26 molecular mechanisms underlying the anti-tumor activity of ATRA in breast-cancer. This suggests that ATRA and immunotherapeutic agents represent rational combinations for the personalized treatment of breast-cancer.
Broad comments:
Overall, this is a very thoroughly conducted study, with sound methodology and reporting. There are minor queries and suggestions mentioned below.
Comments:
- The introduction in my opinion needs to provide more information and build a rationale for the study conducted.
Response: As requested by Reviewer 1, we have substantially re-written and expanded the Introduction of our manuscript (page 2 lines 46-96). Based on our previous studies, we provided further information on the therapeutic potential of ATRA in the personalized treatment of breast cancer. The changes were also introduced to address a similar point raised by Reviewer 2. In addition, we clarified the rationale and scope of the present study.
- Line 52 remove excess spacing
Response: We removed excess spacing as requested.
- Query on whether cell lines authentication has been performed on all 16 cell lines
Response: During the course of the entire study, all the cell lines were authenticated by constantly checking the morphology and the growth doubling time. In addition, all the cell lines were mycoplasma free, as indicated by periodic PCR assays performed on the cell conditioned medium, using the following specific mycoplasma-recognizing primers (forward: 5’TGCACCATCTGTCACTCTGTTAACCTC3’; reverse: 5’ACTCCTACGGGAGGCAGCAGTA3’). This information is now available in the revised version of the Supplementary Methods.
- Can the authors provide a response to why those particular cell lines were chosen for this study, has any work been conducted on primary tumour cells grown from patients?
Response: To address the point raised by the Reviewer, we substantially rewrote the first part of section “2.1. ATRA up-regulates gene-sets controlling interferon/immune-modulatory responses and antigen-presentation in breast-cancer cell-lines” and we provided an explanation for the selection of the 16 cell-lines (page 3, lines 101-126): “…In previous studies, we profiled a large group of over 50 breast-cancer cell-lines for their sensitivity to the anti-proliferative effects of ATRA, using a quantitative index which we denominated “ATRA-score “ [9,10]. To define the gene-sets involved in the growth-inhibitory action of ATRA, we selected a representative panel of 16 cell-lines balanced for the luminal/basal phenotype and for the sensitivity to ATRA (Figure 1A). Indeed, 8 cell-lines have a luminal phenotype, while the other 8 show a basal phenotype and can be further sub-grouped into triple-negative (TNBC) and triple-negative-mesenchymal (TNBC-mes), according to their morphological characteristics and constitutive gene-expression profiles. We re-assessed the sensitivity of the 16 breast-cancer cell-lines to the anti-proliferative effects of ATRA, using a recently developed and optimized version of the ATRA-score [9, 10] (see Materials and Methods). Four luminal cell-lines (SK-BR-3/HCC-1500/CAMA-1/MDA-MB-361) are highly sensitive to ATRA, while the other ones (HCC-202/MDA-MB-175VII/ZR.751/HCC-1419) show relative resistance to the anti-proliferative effects of the retinoid (Figure 1B). However, consistent with the observed higher ATRA-sensitivity of luminal cell-lines relative to the basal counterparts [9], SK-BR-3, HCC-1500, CAMA-1 and MDA-MB-361 cells cluster into the high sensitivity group, while HCC-202, MDA-MB-175VII, ZR.751 and HCC-1419 cells cluster into the intermediate sensitivity group. As for the basal counterparts, 4 cell-lines (HCC-1599/MB-157/MDA-MB-157/Hs578T) can be classified as responsive to ATRA, while the remaining four cell lines (HCC-1187/CAL-851/MDA-MB-436/MDA-MB-231) are highly resistant to the retinoid. Indeed, HCC-1599 cells are endowed with the highest ATRA-score value of the entire panel, while the ATRA-score values aggregate MB-157, MDA-MB-157 and HS578T cells into the intermediate sensitivity group (Figure 1B). In line with the observed resistance to ATRA, the ATRA-score values of HCC-1187, CAL-851, MDA-MB-436 and MDA-MB-231 cells assemble them into the low-sensitivity group. No association is observed between ATRA-sensitivity and the TNBC or TNBC-mes phenotype of the 8 basal cell-lines. In fact, two TNBC (HCC-1599/MB-157) and two TNBC-mes cell-lines (MDA-MB-157/HS578T) are sensitive two ATRA. By the same token, two TNBC (CAL-851/HCC-187) and two TNBC-mes cell-lines (MDA-MB-231/ MDA-MB-436) show resistance to the anti-proliferative effects of the retinoid.”
With respect to the second point of the reviewer, the only experiments performed on primary tumours involve short-term tissue cultures, as described in the manuscript. We have never performed studies in primary tumour cells grown from patients, as it is our experience that it is very difficult to isolate and maintain freshly isolated primary breast cancer cells.
- Did the authors collect any patient clinicopathological details on collected primary tumour samples, if so it would be interesting to view a summary of patient characteristics?
Response: As requested, the clinical characteristics of the 9 patients used for the studies described in Fig. 4D are available in the table added to the Supplementary Methods section of the Supplementary Information file under the heading “Primary breast cancer specimens”. The point is now mentioned in the Materials and Methods section of the manuscript: “4.2. Tissue slice cultures of primary breast cancer specimens” (page 16, lines 595-596).
- Section 4.5 need to provide information on antibody concentrations used for the experiments, overall a more detailed description of this procedures would be beneficial. The same is the case for the description of PCR reaction in the following section. What were the controls?
Response: As requested, the new version of the Materials and Methods section (“4.5. Western blots and FACS analysis”) contains the dilutions of all the antibodies used for the Western blot analyses (page 17, line 630-644). A more detailed description of the PCR methodology used for the determination of the IRF1 and DTX3L mRNAs is now available in the Materials and Methods section under the heading “4.6. Polymerase Chain Reaction” (page 17, lines 645-654). As described in the new version of the manuscript, the internal control used for the normalization of the data is the 18S RNA, which was determined with the use of the standard Taqman gene expression assay, Hs99999901_s1 (ThermoFisher Scientific).

Reviewer 2 Report
This is a very interesting and well-presented study. The authors provide essential and critical evidence to reveal the molecular mechanisms underlying the anti-tumor activity of All-trans retinoic acid (ATRA) in breast-cancer. This manuscript used transcriptomic experiments performed on ATRA-treated breast-cancer cell-lines, short-term tissue cultures of patient-derived mammary-tumors and a xenograft model, discovering that ATRA up-regulates gene-networks involved in interferon-responses, immune-modulation and antigen- presentation in retinoid-sensitive cells and tumors characterized by poor immunogenicity. In addition, ATRA stimulates processes controlling the sensitivity to immuno-modulatory drugs, such as immune-checkpoint-inhibitors, suggesting that ATRA and immunotherapeutic agents represent rational combinations for the personalized treatment of breast-cancer. Again, I think this is a very interesting and well-presented study. I just note two minor issues to point out the potential flaws in this manuscript.
1: Since this study is focusing on breast cancer therapy by using ATRA, some previous studies relate to ATRA treatment on breast cancers should be presented in the introduction sections.
2: In figure 7E, it is easy to know the knockdown efficiency of IRF1 can be compared in the 1, 3 and 5 lands (1:0.2:0). However, it seems no significant difference on DTX3L shRNAs in figure 7F. This maybe the reason that caused a minor anti proliferation effect on shDTX3L, compared to shNC and shVOID. It is suggest the authors to provide the efficient DTX3L knockdown evidence in the revision version.
Author Response
Reviewer 2
This is a very interesting and well-presented study. The authors provide essential and critical evidence to reveal the molecular mechanisms underlying the anti-tumor activity of All-trans retinoic acid (ATRA) in breast-cancer. This manuscript used transcriptomic experiments performed on ATRA-treated breast-cancer cell-lines, short-term tissue cultures of patient-derived mammary-tumors and a xenograft model, discovering that ATRA up-regulates gene-networks involved in interferon-responses, immune-modulation and antigen- presentation in retinoid-sensitive cells and tumors characterized by poor immunogenicity. In addition, ATRA stimulates processes controlling the sensitivity to immuno-modulatory drugs, such as immune-checkpoint-inhibitors, suggesting that ATRA and immunotherapeutic agents represent rational combinations for the personalized treatment of breast-cancer. Again, I think this is a very interesting and well-presented study. I just note two minor issues to point out the potential flaws in this manuscript.
1: Since this study is focusing on breast cancer therapy by using ATRA, some previous studies relate to ATRA treatment on breast cancers should be presented in the introduction sections.
Response: As requested by the Reviewer, the Introduction of the new version of the manuscript is substantially expanded (page 2, lines 46-96). Now, the section contains relevant pieces of information obtained from previous articles that we published on the therapeutic potential of ATRA in the personalized treatment of breast cancer. Please see also the response provided to the first point raised by Reviewer 1.
2: In figure 7E, it is easy to know the knockdown efficiency of IRF1 can be compared in the 1, 3 and 5 lands (1:0.2:0). However, it seems no significant difference on DTX3L shRNAs in figure 7F. This maybe the reason that caused a minor anti proliferation effect on shDTX3L, compared to shNC and shVOID. It is suggest the authors to provide the efficient DTX3L knockdown evidence in the revision version.
Response: The quantitative densitometric data shown in Figure 7F (DTX3L/TUB values) demonstrate that shDTX3L(a) and shDTX3L(b) reduce the expression of the DTX3L protein relative to the shNC and shVOID controls significantly (0.1 vs 0.3, approximately 75% reduction). In addition and as stated, the ATRA-dependent induction of the DTX3L protein observed in shNC and shVOID controls is suppressed in shDTX3L(a) and shDTX3L(b) infected cells (75%-95% reduction). Nevertheless, as suggested by the Reviewer, we added further data obtained at the RNA level by RT-PCR analysis of the DTX3L mRNA which clearly support the protein data. The new data are available in Suppl. Fig. S18. In accordance with these new data, we modified the text of section “2.5. IRF1 and DTX3L knock-down affects the anti-proliferative activity of ATRA in opposite directions” (page 12, lines 465-474), as shown below:
“….Cells expressing shDTX3L(a) and shDTX3L(b) show lower constitutive levels of DTX3L than the shVOID and shNC counterparts (Figure 7F, see DTX3L/TUB values). The silencing action of shDTX3L(a) and shDTX3L(b) is confirmed at the mRNA level, as indicated by the PCR results obtained for the DTX3L transcript which are shown in Supplementary Figure S18. In addition, the ATRA-dependent induction of DTX3L protein and mRNA observed in shVOID and shNC cells is suppressed in the shDTX3L(a) and shDTX3L(b) counterparts (Figure 7F and Supplementary Figure S18). Relative to what is detected with IRF1 silencing, DTX3L knock-down results in functionally opposite effects. In fact, DTX3L knock-down renders SK-BR-3 cells more responsive rather than more refractory to the anti-proliferative effects of ATRA.”
Reviewer 3 Report
In the manuscript ‘All-trans retinoic acid stimulates viral mimicry, interferon responses and antigen presentation in breast-cancer cells,’ Bolis et al. present and analyze RNA-seq data to help explain varying sensitivity of breast cancer cells to ATRA. Gene-set enrichment indicates that interferon regulation is a likely contributor to the changes in gene expression observed in cell lines, and may be associated with the responses to ATRA. This work is supported by RNA and protein expression data. I have reviewed a previous version of this manuscript and appreciate the revisions which have improved this manuscript. I have several remaining comments on this highly relevant study.
Major comments:
- From the Methods, I see that a DESeq2 analysis was performed. The manuscript could be improved by analysis of genes differentially expressed between groups, without necessarily being correlated with ATRA score. This would allow for the possibility that the numeric value of the ATRA score is not the definitive determinant of sensitivity.
- It is not clear to me that significant evidence of viral mimicry is presented in this manuscript. The only data appears in Fig 5B and 5C, and no statistical testing is mentioned, This separation is also distinct from all previous analyses where ordering was done on ATRA-score alone, irrespective of subtype. I appreciate the addition of commentary on how this relates to ATRA score, but it is still not clear to me why this data is separated by subtype.
Minor comments:
- The PCA demonstrates clear distinction between TNBC-basal like and TNBC-mesenchymal like; why is this not maintained throughout the analysis? There is an inconsistent separation by subtype (sometimes PAM50 classes, sometimes just luminal/basal, sometimes including mesenchymal TNBC). Rationale for this should be provided in the text.
- The scores appearing in Figure 1B and used as the basis for all data analysis are not consistent with previous work from this group (Centritto 2015) and are presented without error. For example, MDAMB157 shows a score of ~0.3 in Figure 1B, but has a score >0.7 in Centritto 2015. Without additional context, it is also not clear why HCC1419 is considered “low” (Fig 1B) while it looks to be more closely related to the “intermediate” grouping.
- The correlation between number of genes modulated by ATRA and ATRA score is not a useful analysis and should be removed. (Supp Fig S1B), text reference pg 3 lines 103-104.
- Would like to see cell-line clustering permitted (x axis) as it appears in some cases that the groupings the authors have suggested based on ATRA-score are not necessarily reflected in the gene expression data.
- Fig 6E/F - authors should comment on the significant differences between the two panels (IC50 in F is ~ 1-2x10-9 while in E is much closer to 2x10-8) Would still like to see this.
Author Response
Reviewer 3
In the manuscript ‘All-trans retinoic acid stimulates viral mimicry, interferon responses and antigen presentation in breast-cancer cells,’ Bolis et al. present and analyze RNA-seq data to help explain varying sensitivity of breast cancer cells to ATRA. Gene-set enrichment indicates that interferon regulation is a likely contributor to the changes in gene expression observed in cell lines, and may be associated with the responses to ATRA. This work is supported by RNA and protein expression data. I have reviewed a previous version of this manuscript and appreciate the revisions which have improved this manuscript. I have several remaining comments on this highly relevant study.
Major comments:
- From the Methods, I see that a DESeq2 analysis was performed. The manuscript could be improved by analysis of genes differentially expressed between groups, without necessarily being correlated with ATRA score. This would allow for the possibility that the numeric value of the ATRA score is not the definitive determinant of sensitivity.
Response: According to the suggestion of the Reviewer, we divided the cell-lines in 3 groups characterized by low, intermediate and high sensitivity to ATRA on the basis of the results presented in the corrected version of Fig. 1. We re-analyzed the RNA-seq results to identify the genes up- and down-regulated by ATRA in the three subgroups of cell-lines. The results obtained were subjected to GSEA analysis using the HALLMARK gene-set collection. The results of this analysis are presented in the new versions of Table S3 and Suppl. Fig. S7. The new data are described and discussed in section “2.1. ATRA up-regulates gene-sets controlling interferon/immune-modulatory responses and antigen-presentation in breast-cancer cell-lines” (page 4, lines 169-175 and page 5, lines 176-187), as follows: “…To confirm and extend the data described above, we took a less stringent approach and identified the genes up- and down-regulated by ATRA in cell-lines characterized by low, intermediate and high retinoid-sensitivity (Supplementary Table S3). In the group of cell-lines showing low-sensitivity, ATRA causes significant modifications in the expression of 1110 genes (Up-regulated = 555; down-regulated = 555; adjusted p-value < 0.05, following correction with the Benjamini–Hochberg method). In intermediate- and high-sensitivity cell-lines, the retinoid up-regulates 2844 and 3151 mRNAs, while it down-regulates 3296 and 2938 mRNAs, respectively (adjusted p-value < 0.05). GSEA performed with the HALLMARK gene-set collection demonstrates enrichment of the pathways listed in Supplementary Table S3. For each cell-line group the top pathways showing a threshold FDR (False Discovery Rate) value > 0.05 are illustrated in Supplementary Fig. S7. As expected, ATRA causes a significant down-regulation of the gene-networks involved in cell proliferation (“E2F-Targets” and “G2M-checkpoint”) and MYC activity (“Myc-Targets”) only in cell-lines characterized by intermediate- and high sensitivity to the retinoid. With respect to this last point, it is interesting to notice that ATRA up-regulates the “Myc-Targets” gene-set in the cell-lines characterized by low retinoid sensitivity, supporting the idea that an anti-MYC action underlies the growth-inhibitory effects of the retinoid. By the same token, ATRA-dependent up-regulation of the “Interferon-Alpha-Response” and “Interferon-Gamma-Response” gene-sets is observed in the cell-line groups showing intermediate and high retinoid-sensitivity. Overall, these results confirm the data illustrated in Figure 3A.”.
- It is not clear to me that significant evidence of viral mimicry is presented in this manuscript. The only data appears in Fig 5B and 5C, and no statistical testing is mentioned, This separation is also distinct from all previous analyses where ordering was done on ATRA-score alone, irrespective of subtype. I appreciate the addition of commentary on how this relates to ATRA score, but it is still not clear to me why this data is separated by subtype.
Response: It is our opinion that our study clearly demonstrates that ATRA causes a viral mimicry response in breast cancer cells, as recognized by Reviewers 1 and 2. This effect is directly associated with an anti-proliferative response in basal cell-lines. In luminal cell-lines, the situation is more complicated, as already described and discussed in the original version of the manuscript. This is the reason as to why the data in Fig. 5B are shown following clustering of the cell-lines for the luminal and basal phenotype. With respect to this last point, please notice that the luminal and basal cell-lines are ordered from left to right on the basis of a decreasing ATRA-score value, as in all the other figures. Nevertheless, to avoid any ambiguity, we added the ATRA-score values above Fig.5B. In addition and as requested by the Reviewer, we added the p-values for the statistical analyses performed on the data presented in Fig. 5B. Furthermore the new Suppl. Table S5 contains the ENV RNA expression data obtained by RNA-seq following treatment of each cell-line with vehicle (DMSO) or ATRA. To clarify the entire issue we made substantial modifications to the text of section “2.3. ATRA-dependent stimulation of viral-mimicry in cell-lines” (page 10, lines 349-376; page 11, lines 377-378), as indicated below:
“Viral-mimicry is an emerging intra-cellular process involved in the activation of IFN-dependent [24] and other immune responses [25,26] by certain anti-tumor agents, such as demethylating compounds [27,28]. As ATRA-dependent induction of interferon-responsive genes is accompanied by up-regulation of gene-sets involved in inflammatory/immune responses (see “Inflammatory-Response” in Figure 3A), it is possible that ATRA reactivates endogenous retroviruses (ENVs) resulting in a viral-mimicry response [29]. To support this hypothesis, we re-analyzed the RNA-seq data looking for ENV derived small RNAs. The RNA-seq results permit the identification of more than 5 x 106 ENV derived small RNAs which can be grouped in 42 distinct families (Supplementary Table S5). Consistent with the hypothesis that ATRA activates viral mimicry, the RNA-seq data demonstrate that the retinoid causes a significant up-regulation of various classes of RNAs deriving from ENV transposable-elements in different cell-lines. To quantitate the overall effect exerted by ATRA and to correlate this parameter with retinoid sensitivity, we determined the median induction values of the single classes of ENV-derived mRNAs identified in each cell-line (Figure 5B). If these median induction values and the ATRA-scores are linearly correlated across our entire panel of cell-lines (Supplementary Fig. S13), the calculated correlation index is low (R2 = 0.1629). A close inspection of the diagram suggests that the low R2 value is largely explained by the results obtained in luminal cell-lines, which prompted us to perform the same type of analysis following separation of the luminal and basal cell-lines. In luminal cell-lines, no correlation (R2 = 0.0043) is observed between the ATRA-scores and the median values of the ATRA-dependent induction of these RNAs (Figure 5C). Indeed, HCC-1419 and ZR75.1, which are characterized by the lowest ATRA-scores, show a relatively high ATRA-dependent induction of ENV-derived RNAs. The situation is different in basal cell-lines, where the two parameters are directly correlated (R2 = 0.5788) (Figure 5C). In fact, ATRA up-regulates ENV-derived RNAs only in the 4 retinoid-sensitive basal cell-lines (Figure 5B). Thus, in the luminal cellular context, the ATRA-triggered induction of ENV-derived RNAs and the consequent viral mimicry response are not associated with sensitivity to the anti-proliferative action of the retinoid. In contrast, the two retinoid-dependent processes are activated only in basal cell-lines characterized by sensitivity to the growth inhibitory effects of ATRA. ATRA-dependent induction of ENV-derived RNAs is a relatively early event largely preceding the cell-growth arrest observed in the presence of the retinoid. In fact increased expression of these RNAs is already observed following 8-hour exposure to the retinoid in sensitive HCC-1599 and MB-157 basal cell-lines (data not shown).”.
In addition, we modified chapter 4.8 (Quantification of Transposable elements) of the Materials and Methods section (page 18, lines 675-688) to provide a detailed description of the analyses performed: “Nearly half of the human genome consists of repetitive elements that are tightly regulated to protect the host genome from destructive consequences associated to their inappropriate reactivation [49]. Both full length and fragmented copies of these viral genomes have propagated through host genomes to produce repeating instances of their sequences [50]. To quantify the expression of these transposable elements, we retrieved their genomic positions from the RepeatMasker database (http://www.repeatmasker.org/). We built a custom GTF file (gene transfer file, GTF) including these annotations along with those of canonical genes and quantified their abundance using STAR [51]. To avoid detection of false positives, we discarded all transposable elements that showed any overlap to known gene-associated exons, according to the Gencode annotations [52]. Repetitive elements were grouped into 42 distinct families as reported in Supplementary Table 5. Differential expression of repetitive elements was performed using the DESeq pipeline. In order to obtain a single quantification for all repetitive elements, we further grouped all members of the above mentioned 42 classes into a single meta-gene which consisted of 5482861 elements. Differential expression and statistics were computed as previously described in the above section 4.7.”.
Minor comments:
- The PCA demonstrates clear distinction between TNBC-basal like and TNBC-mesenchymal like; why is this not maintained throughout the analysis? There is an inconsistent separation by subtype (sometimes PAM50 classes, sometimes just luminal/basal, sometimes including mesenchymal TNBC). Rationale for this should be provided in the text.
Response: The separation of basal cell-lines into TNBC and TNBC-mes is due to the different constitutive profiles of gene-expression observed in the two types of breast cancer cells. However, the separation in these two sub-groups has no relevance in terms of the ATRA-sensitivity observed in basal breast cancer cell-lines. In fact, no association is evident between the ATRA-scores determined and the TNBC or TNBC-mes phenotype of the 8 basal cell-lines. With respect to this, it should be noticed that two TNBC cell-lines (HCC-1599/MB-157) and two TNBC-mes cell-lines (MDA-MB-157/HS578T) are sensitive to ATRA. By the same token, two TNBC cell-lines (CAL-851/HCC-187) and two TNBC-mes cell-lines (MDA-MB-231/ MDA-MB-436) show resistance to the anti-proliferative effects of the retinoid. In the new version of the manuscript the point is clarified by extensive re-writing of the first part of section “2.1. ATRA up-regulates gene-sets controlling interferon/immune-modulatory responses and antigen-presentation in breast-cancer cell-lines” (page 3, lines 101-134). Given this and for the sake of simplicity, we used the basal and luminal denomination throughout the remainder of our study. As for the other comment, please notice that we used the PAM50 classification only when we performed analyses on the TCGA dataset. In particular, the PAM50 denomination is present only in Fig.6. We think that the new version of the manuscript clarifies all the ambiguities which were present in its original version.
- The scores appearing in Figure 1B and used as the basis for all data analysis are not consistent with previous work from this group (Centritto 2015) and are presented without error. For example, MDAMB157 shows a score of ~0.3 in Figure 1B, but has a score >0.7 in Centritto 2015. Without additional context, it is also not clear why HCC1419 is considered “low” (Fig 1B) while it looks to be more closely related to the “intermediate” grouping.
Response: Please notice that the slight differences in the ATRA-score values presented in this report and in our previous paper (Centritto 2015) are due to the fact that the methods used for the calculation of the ATRA-score values are slightly different, as now clearly stated under section “4.3 ATRA-score” of the new version of the Materials and Methods section. With respect to this last issue, it is worthwhile mentioning that we have substantially modified the text of this section to clarify the point (page 16, lines 597-605): “…The sensitivity of the cell-lines to the anti-proliferative action of ATRA was determined following optimization of the method necessary to calculate the ATRA-score quantitative index [9]. Briefly, breast-cancer cell-lines were exposed to vehicle (DMSO) or five logarithmically increasing concentrations of ATRA (0.001–10.0 mM) for 9 days. Cell growth was determined with the sulforhodamine assay [9]. Each experimental point consisted of 6 replicates. For each cell line at least two independent experiments were carried out. ATRA-score = log2 transformation of the product of AUC X Amax, rescaled in a range between 0 and 1. “0” and “1” indicate total resistance and maximal sensitivity to ATRA, respectively.”. Please notice that we did not add any error bar for the data, as the ATRA-scores were calculated on 6 replicates and the standard deviation for the values is below 10%. As for HCC1419, we agree with the Reviewer that HCC-1419 should be considered as characterized by intermediate sensitivity to ATRA. For this reason, we lowered the intermediate threshold value line below the HCC-1419 cell line in the new version of Fig. 1B. To comply with the positioning of HCC-1419 cells within the intermediate ATRA-sensitivity group we made a small change to the text (page 3, lines 114-116).
- The correlation between number of genes modulated by ATRA and ATRA score is not a useful analysis and should be removed. (Supp Fig S1B), text reference pg 3 lines 103-104.
Response: As suggested by the reviewer, we removed the correlation graph in the inset of Suppl. Fig. S1B, as we agree that these data are not particularly useful. To comply with this, we changed the text of the manuscript (page 3, lines 133-134): “…ATRA treatment does not cause transitions across the 3 groups, although the retinoid up- and down-regulates several genes in each cell-line, (Supplementary Figure S1B).”.
- Would like to see cell-line clustering permitted (x axis) as it appears in some cases that the groupings the authors have suggested based on ATRA-score are not necessarily reflected in the gene expression data.
Response: We think that the Reviewer agrees with us that the gene-expression heat-maps presented in Fig. 2, Supplementary Fig. S1, Supplementary Fig. S4 and Supplementary Fig. S5 show patterns that are reasonably correlated with the ATRA-scores calculated for each cell-line. Thus we think that the type of clustering adopted represents the best option to be used for the illustration of the data.
- Fig 6E/F - authors should comment on the significant differences between the two panels (IC50 in F is ~ 1-2x10-9 while in E is much closer to 2x10-8) Would still like to see this.
Response: We think that the Reviewer refers to Fig. 7E and 7F and not to Fig. 6E/F, as stated. Please notice that the data obtained in Fig.7E with the siRNAs were obtained following transient transfection and refer to the whole cellular population. In contrast the data in Fig.7D were obtained following stable infection of the shRNA constructs and antibiotic selection of the over-expressing clones. Clearly, the two experimental conditions are completely different, which is likely to explain the differences in the IC50 values observed. Thus, the absolute IC50 values obtained with ATRA in the corresponding controls (siNC and shVOID/shNC) cannot be directly compared. The only curves that can be compared are siNC versus siIRF1(a) or siIRF1(b) and shVOID/shNC versus shDTX3l(a)/shDTX3l(b), as we did in the manuscript. Given the possible confounding factors associated, we think that the Reviewer would agree that the differences in IC50 values should not be emphasized, because the point is of minor interest. In addition the point does not alter the conclusions drawn from these functional studies.
Round 2
Reviewer 3 Report
I thank the authors for their considered responses to my review. I am satisfied with the revisions and appreciate the improved manuscript.